# ROBOTOUILLE: AN ASYNCHRONOUS PLANNING BENCHMARK FOR LLM AGENTS

**Gonzalo Gonzalez-Pumariega**[*]**, Leong Su Yean, Neha Sunkara, Sanjiban Choudhury**
Cornell University

## ABSTRACT

Effective asynchronous planning, or the ability to efficiently reason and plan over states and actions that must happen in parallel or sequentially, is essential for agents that must account for time delays, reason over diverse long-horizon tasks, and collaborate with other agents. While large language model (LLM) agents show promise in high-level task planning, current benchmarks focus primarily on short-horizon tasks and do not evaluate such asynchronous planning capabilities. We introduce ROBOTOUILLE, a challenging benchmark environment designed to test LLM agents' ability to handle long-horizon asynchronous scenarios. Our synchronous and asynchronous datasets capture increasingly complex planning challenges that go beyond existing benchmarks, requiring agents to manage overlapping tasks and interruptions. Our results show that `ReAct` (`gpt4-o`) achieves 47% on synchronous tasks but only 11% on asynchronous tasks, highlighting significant room for improvement. We further analyze failure modes, demonstrating the need for LLM agents to better incorporate long-horizon feedback and self-audit their reasoning during task execution. Code is available here.

## 1 INTRODUCTION

Large language models (LLMs) have demonstrated impressive reasoning and task planning capabilities in short-horizon single-agent environments with clearly defined sequential tasks (Yao et al., 2022; 2023b; Shinn et al., 2023); however, decision-making in the real world introduces a more intricate array of challenges. Consider an assistant that helps you with cooking a recipe. It must be able to handle (1) *time delays* such as boiling spaghetti, which takes time to complete. An efficient agent would move onto other steps instead of waiting for the spaghetti to fully cook. It should also handle (2) *diverse long-horizon tasks* that require the assistant to satisfy multiple objectives and reason about dependencies between different actions. Finally, the assistant should handle (3) *multiple agents* by coordinating with others or distributing tasks based on each agent's capability. To tackle these challenges, an agent must be capable of **asynchronous planning**, or the ability to efficiently reason and plan over states and actions that must happen in parallel or sequentially. With this capability, an agent can coordinate time delays, break down long horizon tasks into subtasks, and efficiently assign subtasks to multiple agents.

To improve asynchronous planning capability, we are interested in a benchmark (Table 1) that stress tests LLM agents using time delays. AsyncHow Lin et al. (2024) benchmarks asynchronous planning but does not use an interactive environment, lacking support for closed-loop planning agents. ALFWorld (Shridhar et al., 2021), WebShop (Yao et al., 2023a) and PlanBench (Valmeekam et al., 2023b) offer long-horizon diverse tasks (up to 50, 48 and 90 steps respectively) but evaluate with a single agent and no time delays. VirtualHome (Puig et al., 2018) offers long-horizon (up to 96 steps) and multi-agent tasks with procedural generation for extra diversity but also lacks time delays.

To address these gaps, we introduce ROBOTOUILLE, a simulator for cooking diverse recipes designed to stress test LLM agents (Figure 1). ROBOTOUILLE tests asynchronous planning through tasks that take time like cooking meat for burgers or sandwiches or filling up a pot with water to cook soup. Its fully customizable JSON backend allows for the addition of new states, actions, and goals simplifying the creation of diverse long-horizon tasks. Finally, ROBOTOUILLE supports turn-based and real-time multi-agent execution either locally or on the network.

---

[*]Corresponding author. Email: gg387@cornell.edu

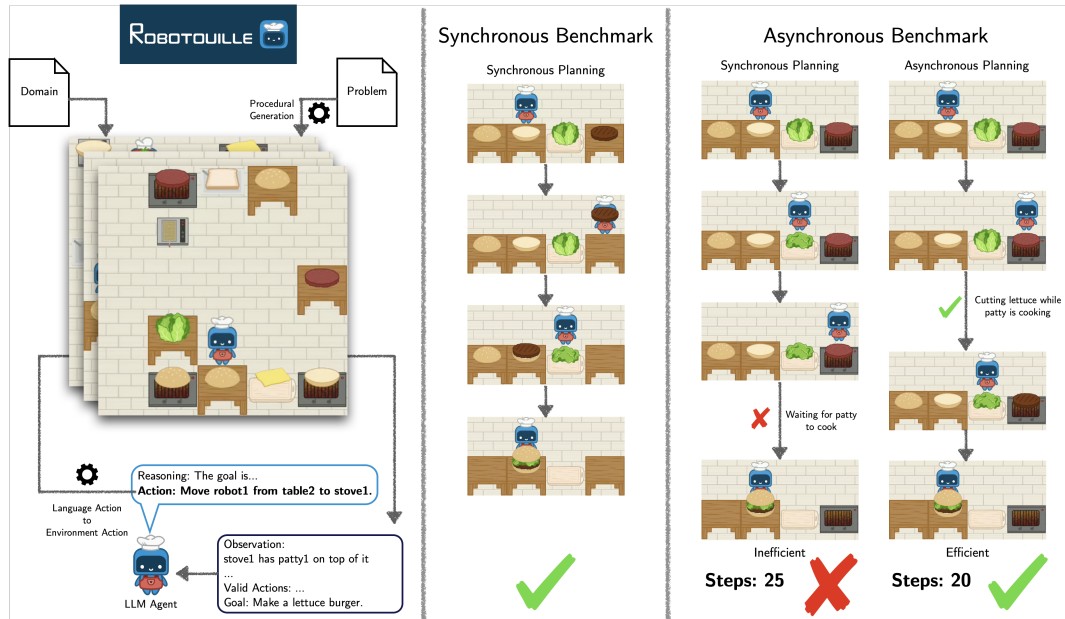

Figure 1: Overview of ROBOTOUILLE along with examples of our synchronous and asynchronous benchmarks. ROBOTOUILLE takes a domain and problem JSON to procedurally generate an environment for an LLM agent to plan in. In the synchronous benchmark, the order that the burger is assembled has minimal impact in the efficiency of the plan. In the asynchronous benchmark, ordering matters due to time delays; leaving the patty to cook before cutting the lettuce is more efficient than leaving the patty to cook after cutting the lett.

In addition, we provide 3 datasets to test LLM agents' synchronous, asynchronous, and multi-agent planning capabilities. We implement 3 baselines for benchmarking the synchronous and asynchronous datasets, leaving multi-agent for future work, and provide analyses on the failure modes to provide insights for future work. Our hope is for the research community to engage with ROBOTOUILLE to create an ecosystem of environments and methods that increase the diversity of our testbed and the capabilities of LLM agents.

Our key contributions include the following

1. We present a new environment, ROBOTOUILLE, for stress testing LLM agents' ability to perform asynchronous planning to handle time delays, diverse long-horizon tasks, and multi-agent.

2. We curate 3 datasets for synchronous, asynchronous, and multi-agent settings, each containing 10 unique tasks each with 10 procedurally generated instances.

3. We implement various LLM baselines, evaluate them on the synchronous and asynchronous datasets, and provide quantitative and qualitative analyses on failure modes.

## 2 ROBOTOUILLE

We formalize ROBOTOUILLE tasks as an MDP with time-delayed effects, $\mathcal{M} =< \mathcal{S}, \mathcal{A}, \mathcal{T}, \mathcal{R} >$. Each state $s \in \mathcal{S}$ is $s = (\hat{s}_t, H_t)$ where $\hat{s}_t$ represents observable state elements like objects or predicates such as `iscut(lettuce1)`, or "`lettuce1` is cut", and `on(lettuce1,table2)`, or "`lettuce1` is on `table2`", and $H_t$ is a set of timer variables $h \in H_t$ each created by actions with a countdown function $h(x) = d - (x - i)$ where $d$ is a delay constant and $i$ is the timer's activation step. Action $a \in \mathcal{A}$ is a grounded action such as `move(robot1, table1, table2)`, or "Move `robot1` from `table1` to `table2`" that may introduce new timers $h$. Actions have preconditions over state predicates which must be met to be valid. For a given state $s$ and action $a$, the transition function $\mathcal{T} \colon \mathcal{S} \times \mathcal{A} \to \mathcal{S}$ returns the next state $s' = (\hat{s}_{t+1}, H_{t+1})$ if $a$ is valid or the current state $s$ if $a$ is invalid. For a valid action step, $\hat{s}_{t+1} = \hat{s}_t \cup \{predicates(h) | h \in H_t, h(t) = 0\}$ to removes expired timers and $H_{t+1} = (H_t - \{h | h(t) = 0\}) \cup \{h | a \text{ adds delay}\}$ to update active timers. The

reward function $\mathcal{R}: \mathcal{S} \to \{0, 1\}$ defines the goal of a given task where for goal state $s_g$, $r(s_g) = 1$. We provide a complexity analysis between synchronous and asynchronous settings in Appendix A.9.

| Benchmark | High-Level Actions | Multi-agent | Procedural Level Generation | Time Delays | Number of Tasks | Longest Plan Horizon |
|---|---|---|---|---|---|---|
| ALFWorld (Shridhar et al., 2021) | ✓ | ✗ | ✗ | ✗ | 3827 | 50 |
| CuisineWorld (Gong et al., 2023) | ✓ | ✓ | ✓ | ✗ | 33 | 11 |
| MiniWoB++ (Liu et al., 2018) | ✓ | ✗ | ✗ | ✗ | 40 | 13 |
| Overcooked-AI (Carroll et al., 2020) | ✗ | ✓ | ✗ | ✓ | 1 | 100 |
| PlanBench (Valmeekam et al., 2023b) | ✓ | ✗ | ✓ | ✗ | 885 | 48 |
| $\tau$-bench (Yao et al., 2024) | ✓ | ✗ | ✓ | ✗ | 165 | 30 |
| WebArena (Zhou et al., 2024) | ✓ | ✗ | ✓ | ✗ | 812 | 30 |
| WebShop (Yao et al., 2023a) | ✓ | ✗ | ✗ | ✗ | 12087 | 90 |
| AgentBench (Liu et al., 2023d) | ✓ | ✓ | ✗ | ✗ | 8 | 35 |
| ARA (Kinniment et al., 2024) | ✓ | ✗ | ✗ | ✗ | 12 | 4 |
| AsyncHow (Lin et al., 2024) | ✓ | ✗ | ✗ | ✓ | 1600 | 9 |
| MAgIC (Xu et al., 2023) | ✓ | ✓ | ✗ | ✗ | 5 | 20 |
| T-Eval (Chen et al., 2024) | ✓ | ✓ | ✗ | ✗ | 23305 | 19 |
| MLAgentBench (Huang et al., 2024) | ✓ | ✗ | ✗ | ✗ | 13 | 50 |
| GAIA (Mialon et al., 2023) | ✓ | ✗ | ✗ | ✗ | 466 | 45 |
| VirtualHome (Puig et al., 2018) | ✓ | ✓ | ✓ | ✗ | 2821 | 96 |
| ROBOTOUILLE (Ours) | ✓ | ✓ | ✓ | ✓ | 30 | 82 |

Table 1: Comparison between ROBOTOUILLE and other benchmarks. See Appendix A.1 for more details.

**Domain and Problem JSONs** ROBOTOUILLE uses JSONs to fully describe a task $\mathcal{M}$ using a domain $\mathcal{D} = <\mathcal{O}_\mathcal{D}, \mathcal{P}_\mathcal{D}, \mathcal{A}_\mathcal{D}>$ and problems $\mathcal{P} = <\mathcal{O}_\mathcal{P}, \mathcal{I}_\mathcal{P}, \mathcal{G}_\mathcal{P}>$, inspired by PDDL (Aeronautiques et al., 1998) and described in Figure 2 (a-b). Domain $\mathcal{D}$ defines the possible states and actions of an environment with object types $\mathcal{O}_\mathcal{D}$, predicate definition $\mathcal{P}_\mathcal{D}$ and action definitions $\mathcal{A}_\mathcal{D}$. Problem $\mathcal{P}$ grounds the domain definitions with objects $\mathcal{O}_\mathcal{P}$, initial state predicates $\mathcal{I}_\mathcal{P}$, and goal $\mathcal{G}_\mathcal{P}$. In addition, $\mathcal{P}_\mathcal{D}$, $\mathcal{A}_\mathcal{D}$ and $\mathcal{G}_\mathcal{P}$ have language representations for an LLM agent.

**Action Effects** We adopt immediate effects from PDDL, where $\mathcal{T}(s, a) = s'$ and $s'$ results from predicates being added or removed due to $a$. To extend actions beyond immediate effects, we introduce **special effects**, which are custom code blocks that allow for complex interactions, such as delayed effects in cooking where predicates are added after a delay. Figure 2 (c) shows an example of a special effect for the cook action. A conditional effect applies the `iscooking` predicate if an item `i1` is on station `s1` and removes it otherwise. In addition, a delayed effect is nested that adds predicate `iscooked(i1)` after a delay specified in the problem JSON (see Appendix A.2).

**Language Goal** Language goals are inherently ambiguous and many states may satisfy them. For example, in Figure 2 (d), the goal `Make lettuce cheese sandwich on table` lacks information about which ingredients or tables to use (in the case where there are multiple) and doesn't specify whether the lettuce is above or below the cheese. We created a flexible goal specification system that captures a combinatorial number of goal states that may satisfy a vague language goal. In this example, by specifying that (1) one bread slice must be directly on the table, (2) another is somewhere at the table while being clear on top and (3) lettuce and cheese must be somewhere at the table, we fully capture all possible outcomes that satisfy the language goal.

**Procedural Generation** ROBOTOUILLE provides procedural generation which works off an existing problem JSON. To ensure that goals can be satisfied, the problem JSON should contain the minimum number of objects that satisfy the goal. The procedural generator shuffles existing objects and adds new objects which allows for stress testing on diverse environments with varying language descriptions and optimal paths to the goal.

**Multi-agent** ROBOTOUILLE supports multi-agent environments by simply adding more players into the problem JSON. These environments can be either turn-based, where an LLM agent controls a single agent at a time, or real-time, where an LLM agent controls all agents simultaneously. We additionally implement networked multi-agent to allow data-collection of human-human play and evaluating agents against humans.

```
"predicate_defs": [{
  "name": "istable",
  "param_types": ["station"],
  "language_descriptors": {
    "0": "{0} is a table"
  }},{
  "name": "item_on",
  "param_types": ["item", "station"],
  "language_descriptors": {
    "0": "{0} is directly on top of {1}",
    "1":
    ↪   "{1} has {0} directly on top of it"
}},...]
```

*(a) Predicate Definitions*

```
"name": "move",
"precons": [{
  "predicate": "loc",
  "params": ["p1", "s1"],
  "is_true": true
}, ...],
"immediate_fx": [{
  "predicate": "loc",
  "params": ["p1", "s2"],
  "is_true": true
}, {
  "predicate": "loc",
  "params": ["p1", "s1"],
  "is_true": false
}, ...],
"sfx": [],
"language_description":
↪   "Move {p1} from {s1} to {s2}"
```

*(b) Action Definitions*

```
"sfx": [{
  "type": "conditional",
  "param": "i1",
  "conditions": [{
    "predicate": "item_on",
    "params": ["i1", "s1"],
    "is_true": true
  }],
  "fx": [{
    "predicate": "iscooking",
    "params": ["i1"],
    "is_true": true
  }],
  "sfx": [{
    "type": "delayed",
    "param": "i1",
    "fx": [{
      "predicate": "iscooked",
      "params": ["i1"],
      "is_true": true
    }, {
      "predicate": "iscooking",
      "params": ["i1"],
      "is_true": false
    }],
    "sfx": []
  }]
}]
```

*(c) Nested special effects for 'cook' action*

```
"goal_description":
↪   "Make lettuce cheese sandwich on table",
"goal": [{
  "predicate": "item_on",
  "args": ["bread", "table"],
  "ids": [1, 2]
}, {
  "predicate": "item_at",
  "args": ["lettuce", "table"],
  "ids": [3, 2]
}, {
  "predicate": "item_at",
  "args": ["cheese", "table"],
  "ids": [4, 2]
}, {
  "predicate": "item_at",
  "args": ["bread", "table"],
  "ids": [5, 2]
}, {
  "predicate": "clear",
  "args": ["bread"],
  "ids": [5]
}]
```

*(d) Goal Description*

Figure 2: ROBOTOUILLE uses domain and problem JSONs to define the MDP and language description of an environment and tasks using (a) predicate definitions, (b) action definitions, (c) special action effects and (d) goal definitions. See Appendix A.2 for other JSONs used.

## 3   DATASET DETAILS

In this section we discuss the contents of the synchronous and asynchronous datasets and their differences. We provide discussion of the in-context example tasks and multi-agent dataset in Appendix A.3. Each dataset contains 10 unique tasks and has 10 procedurally generated instances. Table 3 and Appendix A.6 include visual representations of the tasks and dependency graphs respectively.

**Synchronous Dataset** This dataset consists of tasks involving assembling sandwiches and burgers with ingredients that may need to be cut. Any ingredients that can be cooked are initialized as cooked. Tasks 1 to 3 involve assembling sandwiches of increasing difficulty where Task 1 only involves assembling and Task 2 and 3 involve cutting ingredients. Tasks 4 to 7 involve assembling burgers which differ from sandwiches in that the burger buns have ordering constraints with distinct buns that go on the top and the bottom. Unlike other tasks, Task 6 enforces a strict ordering constraint on the placement of all ingredients. Finally, Tasks 8 to 10 involve the preparation of 2 recipes which increase in difficulty from identical sandwiches, identical burgers, and finally a sandwich and burger with different ingredients.

**Asynchronous Dataset** This dataset consists of tasks including sandwiches and burgers from before but also fried recipes and soup. Unlike the synchronous dataset, ingredients that can be cooked are initialized as uncooked; this allows for asynchronous planning. Tasks 1 to 3 use the same ingredients as those in the synchronous setting except for an added ingredient which must be cooked or fried. We studied these tasks in Appendix A.17 for a closer one-to-one comparison with synchronous tasks and found that asynchronous tasks are more difficult. Tasks 4 and 5 involve making a burger and a fried recipe; Task 4 includes french fries which requires cutting a potato then frying while Task 5 includes fried onions which is the same process with an onion. Tasks 6 to 7 introduce a new recipe, soup, which involves filling a pot with water from a sink, boiling the water, putting ingredients inside, and finally serving in a bowl. Of these subtasks, filling a pot with water and boiling the water are

steps that can be done asynchronously with other tasks. Finally, Tasks 8 to 10 involve making soup along with increasing numbers of sandwiches and burgers.

## 4 EXPERIMENTS

### 4.1 BASELINES

We evaluate LLMs on ROBOTOUILLE using the following baselines: `I/O`, `I/O CoT`, and `ReAct`. `I/O` takes as input the initial state, including valid actions and goal, and outputs an plan directly. `I/O CoT` (Wei et al., 2023) also takes as input the initial state but outputs a plan with chain of thought before each action that estimates the resulting state. Instead of outputting the entire plan, `ReAct` (Yao et al., 2022) outputs reasoning and the next action given the current state, and receives the next state before repeating. We use an ablated version of `ReAct` that only keeps the reasoning and action of the previous timestep in context (along with the base prompt and in-context examples); the improved performance and cost-effectiveness is detailed in Appendix A.7. Each baseline receives a single in-context example on a training example excluded from the testing set. We use temperature 0.7 for all models. All prompts and few-shot examples are located in our codebase here.

### 4.2 RESULTS AND ANALYSIS

#### 4.2.1 OVERALL TAKEAWAYS

- **Closed-loop agents are superior**: The best baseline, `gpt4-o ReAct`, achieves 47% on the synchronous dataset and 11% on the asynchronous dataset, surpassing open-loop approaches `I/O` and `I/O CoT` (Finding 1, Sec 4.2.2).

- **Poor feedback incorporation leads to decreased asynchronous performance**: Despite being closed-loop, `gpt4-o ReAct` failures often make little progress towards the goal (Finding 3, Sec 4.2.2) due to poor failure recovery (Finding 5, Sec 4.2.3). We find that boosting priors improves performance (Finding 7, Sec 4.2.4) but discuss better feedback methods in Section 5.

- **Synchronous and asynchronous failures are closely related**: Both synchronous and asynchronous failures are dominated by rule violations and goal misinterpretation (Finding 4, Sec 4.2.3). We hypothesize that this is due to poor failure recovery (Finding 5, Sec 4.2.3) and agents that recover efficiently could boost performance in both settings.

- **Task prioritization is critical in asynchronous planning**: Proper prioritization of subtasks in asynchronous settings significantly boosts performance (Finding 6, Sec 4.2.4).

#### 4.2.2 SUCCESS AND OPTIMALITY

**Finding 1.** Closed-loop baselines outperform open-loop baselines.

Table 2 shows the success rates of various LLMs baselines on the synchronous and asynchronous datasets. Table 3 shows the task-specific success rates of baselines using `gpt4-o`. Success rate is determined by reaching the goal within 1.5 times the optimal number of steps for the given instance. Baselines exceeding this step limit are terminated.

Among all the LLM baselines, `ReAct` with the `gpt4-o` model performs the best on the synchronous and asynchronous datasets. `I/O` performs worst for most LLMs while `I/O CoT` improves performance. We qualitatively observed `gemini-1.5-flash` failing with `ReAct` since it attempts to solve the few-shot example goal rather than the current environment goal. This is likely due to the long context examples, aligning with findings in Liu et al. (2023c) where LLMs struggle with simple tasks in long contexts. We investigate PLaG (BaG) Lin et al. (2024) and Reflexion Shinn et al. (2023) performance on the asynchronous dataset in Appendix A.15 and achieved small performance improvements. We present the performance of state-of-the-art open source LLMs in Appendix A.16.

When considering task-specific success over `gpt4-o` baselines, `ReAct` generally achieves higher performance per task. While we list the horizon length as a crude difficulty metric, it is evident that success rate is not solely dependent on it. We investigate this further in Appendix A.8. We also investigate different agent failure modes in more depth in Section 4.2.3.

**Finding 2**. Asynchronous successes are less optimal than synchronous ones.

| | Synchronous (%) | | | Asynchronous (%) | | |
|---|---|---|---|---|---|---|
| | I/O | I/O CoT | ReAct | I/O | I/O CoT | ReAct |
| gpt4-o | 4.00 | 14.0 | **47.0** | 1.00 | 1.00 | **11.0** |
| gpt-4o-mini | 4.00 | 10.0 | 11.0 | 0.00 | 1.00 | 0.00 |
| gemini-1.5-flash | 0.00 | 13.0 | 0.00 | 0.00 | 0.00 | 0.00 |
| claude-3-haiku | 1.00 | 2.00 | 2.00 | 0.00 | 0.00 | 0.00 |

Table 2: Success rates of state-of-the-art LLMs on the synchronous and asynchronous datasets.

| | I/O | I/O CoT | ReAct | Horizon Length |
|---|---|---|---|---|
| **Synchronous (%)** | | | | |
| [1] | 20.0 | 40.0 | **70.0** | 10 |
| [2] | 0.00 | 20.0 | **80.0** | 14 |
| [3] | 10.0 | 30.0 | **80.0** | 24 |
| [4] | 0.00 | 10.0 | **40.0** | 10 |
| [5] | 0.00 | 0.00 | **60.0** | 15 |
| [6] | 10.0 | **20.0** | **20.0** | 23 |
| [7] | 0.00 | 0.00 | **50.0** | 36 |
| [8] | 0.00 | 10.0 | **30.0** | 44 |
| [9] | 0.00 | 10.0 | **20.0** | 63 |
| [10] | 0.00 | 0.00 | **20.0** | 57 |
| **Total** | 4.00 | 14.0 | **47.0** | |
| **Asynchronous (%)** | | | | |
| [1] | 10.0 | 0.00 | **20.0** | 21 |
| [2] | 0.00 | 0.00 | **30.0** | 27 |
| [3] | 0.00 | 0.00 | **40.0** | 37 |
| [4] | 0.00 | 0.00 | **10.0** | 42 |
| [5] | 0.00 | **10.0** | 0.00 | 46 |
| [6] | 0.00 | 0.00 | **10.0** | 19 |
| [7] | 0.00 | 0.00 | 0.00 | 42 |
| [8] | 0.00 | 0.00 | 0.00 | 46 |
| [9] | 0.00 | 0.00 | 0.00 | 68 |
| [10] | 0.00 | 0.00 | 0.00 | 82 |
| **Total** | 1.00 | 1.00 | **11.0** | |

Table 3: gpt4-o performance on the synchronous and asynchronous datasets.

Fig. 3 shows a histogram of the binned optimality rates on the successful runs of gpt4-o ReAct on the synchronous and asynchronous datasets. Optimality rate is $\frac{\|\hat{\tau}\|}{\|\tau^*\|}$ where $\|\hat{\tau}\|$ is the number of steps taken by an agent and $\|\tau^*\|$ is the optimal number of steps to reach the goal from the initial state.

For the synchronous dataset, 55.3% of successful attempts are optimal compared to the asynchronous dataset where only 9.1% of successful attempts are optimal. We expect this since the order that tasks are done in the synchronous setting does not affect optimality compared to the asynchronous setting. We also see for the asynchronous dataset that 63.6% of successful attempts are suboptimal in the $(1, 1.25]$ bucket. We qualitatively observe that while the LLM agent usually prioritizes asynchronous subtasks, suboptimal runs were due to inefficient actions, such as waiting while cooking. We further investigate the agent's subtask prioritization in Section 4.2.4.

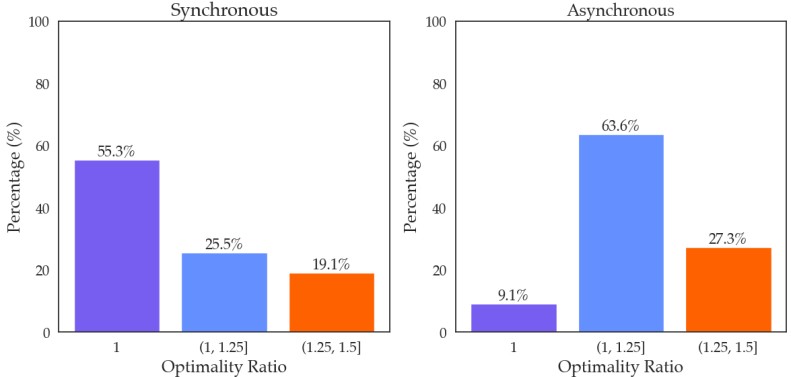

Figure 3: Histogram of the optimality rate for `gpt4-o ReAct` successes on the synchronous and asynchronous datasets. The 1 bin includes attempts that were optimal. Attempts between $(1, 1.5]$ are suboptimal but classified as successful. Attempts greater than an optimality rate of 1.5 are classified as failures.

**Finding 3.** Asynchronous failures make little progress toward the goal.

Fig. 4 shows a histogram of the binned normalized steps to go on the failed runs of `gpt4-o ReAct` on the synchronous and asynchronous datasets. Steps to go is $\frac{\|\tau_{\text{left}}^*\|}{\|\tau^*\|}$ where $\|\tau_{\text{left}}^*\|$ are the optimal number of steps left to reach the goal from the final state in a failed run and normalization factor $\|\tau^*\|$ is the optimal number of steps to reach the goal from the initial state.

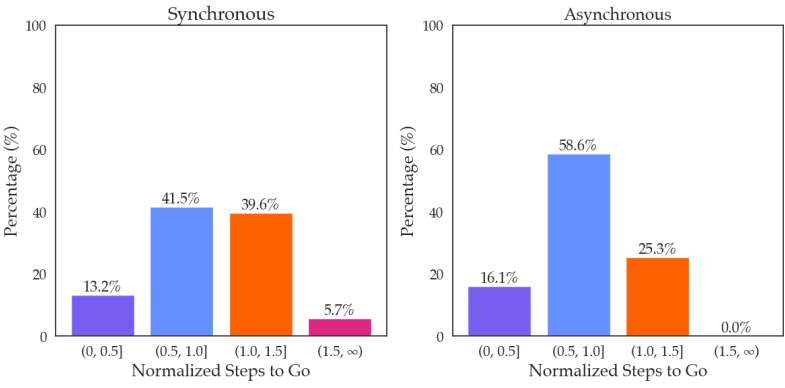

Figure 4: Histogram of the normalized steps to go for `gpt4-o ReAct` failures on the synchronous and asynchronous datasets. The 0 to 0.5 bucket includes attempts that were making progress towards the goal while the 0.5 to 1 bucket includes attempts that made little to no progress towards the goal. Buckets greater than 1 includes attempts that traversed further away from the goal.

For the asynchronous dataset, about 58.6% of the failures are in the $(0.5, 1.0]$ bucket, which shows that most attempts made little to no progress towards the goal. We also see this in the synchronous dataset, with 41.5% of failures in the $(0.5, 1.0]$ bucket. We show quantitative results of `gpt4-o ReAct`'s ineffective failure recovery in Section 4.2.3 suggesting that failures on the asynchronous dataset are mainly due to little progress being made. In contrast, we see 45.3% failures in the

synchronous dataset from $(1.0, \infty)$ which show that most attempts make progress away from the goal. The asynchronous dataset only has 25.3% failures from $(1.0, \infty)$. We present qualitatively annotated failures in Section 4.2.3 that suggest failures on the synchronous dataset are due to misunderstanding the goal.

### 4.2.3 FAILURE MODE ANALYSIS

**Finding 4.** Dominant failures in both settings stem from rule violations and goal misinterpretations.

Fig. 5 shows a nested piechart that captures failure modes of `gpt4-o ReAct` on the synchronous and asynchronous datasets. We define our failure modes in terms of uncertainty over the MDP of the environment. The 4 main failure categories include uncertainty in the state (S), actions (A), transition function (T) and the goal (G). For a detailed description of the subcategories and dataset annotation, see Appendix A.10.

For synchronous failures, the uncertainty in the goal accounts for the majority at 64.1% followed by the uncertainty in the transition function at 32.1%. Goal failures could be due to (1) an incorrect understanding at the start of the plan or (2) a mistake during plan execution, such as using an ingredient without cutting it, which is incorrectly believed to satisfy the goal. We observe that case (1) occurs 28.3% of the time under Bad Start; the LLM agent restates goals incorrectly for complex tasks with strict ordering dependencies like Task 6 or tasks with many diverse ingredients like Task 10 which we show in Appendix A.12. We observe that case (2) occurs 35.8% of the time under the remaining subcategories; although the LLM agent starts with a correct goal, it misunderstands the goal during execution by choosing the wrong action. For transition failures, violating the 'one item at a station' rule accounts for the majority of failures at 24.5%. We qualitatively observe that the agent attempts to use cutting stations for ingredient preparation while other items occupy the station; however, we also observe that once the agent has recovered from this failure it is unlikely to repeat it which we show in Appendix A.14.

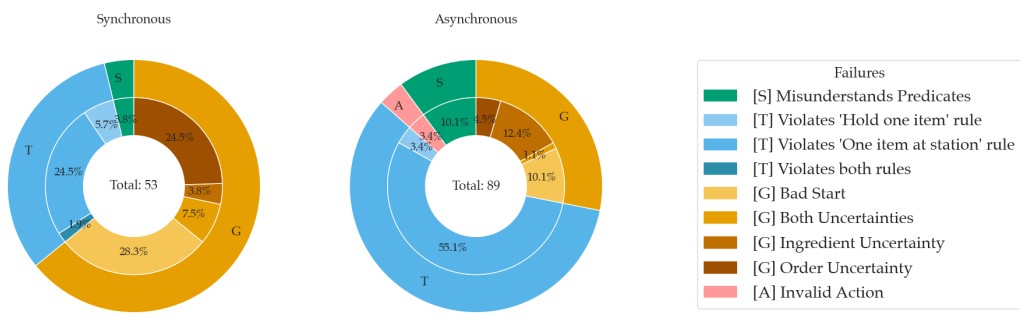

Figure 5: Nested pie chart of `gpt4-o ReAct` failure modes capturing uncertainties in the MDP. The main categories are on the outer circle representing the uncertainty in the state space (S), action space (A), transition function (T), or reward/goal (G). The subcategories on the inner circle represent the dominant cause of failure and are described further in Appendix A.10.

For asynchronous failures, the inverse is true with uncertainty in the transition function accounting for 56.8% of failures and uncertainty in the goal accounting for 34.1% of failures. Similar to the synchronous failures, violating the 'one item at a station' rule dominates failures at 53.4%. This is due to the increased number of unique stations in the asynchronous setting compared to the synchronous setting which increases the potential number of recoveries necessary. In the synchronous setting, which only uses the cutting board station, an agent may need to recover once from violating the 'one item at a station' rule. In the asynchronous setting, which uses stoves, fryers, and sinks, an agent, in the worst case, may need to recover from violating rules on each station in a task.

We point out that while we designed the synchronous and asynchronous datasets to test different capabilities of LLM agents, we mainly observe similar transition failures in both settings. This demonstrates the need to improve LLM agents at following environment constraints to improve their decision-making ability. We investigate this further in Section 4.2.4.

**Finding 5.** Asynchronous recovery is worse than synchronous recovery.

Fig. 6 shows a histogram of the repeated transitions of `gpt4-o ReAct` runs on the synchronous and asynchronous datasets. We use repeated transitions as a proxy for measuring `ReAct`'s effectiveness at recovering from failure.

In both the synchronous and asynchronous datasets, we see that the majority of successes have 0 repeated transitions; few successes have repeated transitions but successfully recover. For failures, the asynchronous dataset's lower and upper quartiles are 103.1% and 55.8% larger than the synchronous dataset's quartiles. This means that failures on the asynchronous dataset are expected to have higher repeated transitions; this ineffectiveness at recovery aligns with the transition failures being dominant for the asynchronous setting in Fig. 5. Similarly, since the synchronous dataset has lower quartiles than the asynchronous dataset, we expect to see less repeated transitions which suggests less transition failures. We do a further investigation in Appendix A.13 by introducing stochasticity and find that even in synchronous settings LLMs struggle at recovery.

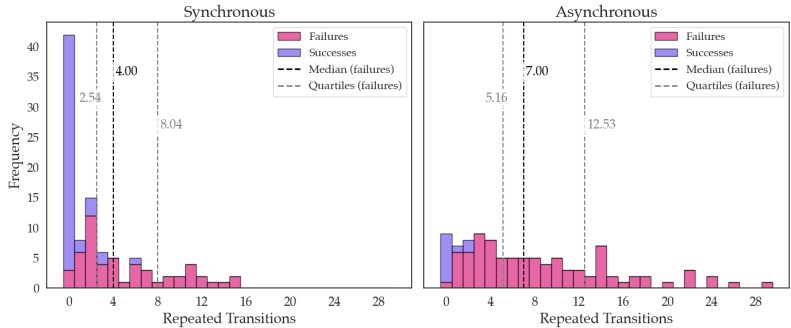

Figure 6: Histogram of the repeated transitions of `gpt4-o ReAct` runs on the synchronous and asynchronous datasets. The median and quartiles of the asynchronous dataset are generally higher than those of the synchronous dataset, indicating higher repeated transitions.

### 4.2.4 FOLLOW-UP INVESTIGATION

From the previous experiments, we conclude that LLM agents struggle in the asynchronous dataset due to simple failures that arise in the synchronous dataset. In order to have a better understanding of how to improve LLM agent capabilities on asynchronous planning, we look into asynchronous subtask prioritization and boosting performance.

**Finding 6.** Proper asynchronous prioritization boosts performance.

Efficient asynchronous planning requires prioritizing subtasks that can be performed asynchronously. We investigate how success rate changes with asynchronous task prioritization to understand the impact of asynchronous planning on the results. Our hypothesis is that prioritizing asynchronous subtasks leads to higher success rates because the planned trajectory is shorter and reaches the goal within the maximum step limit. We find that the success rate conditioned on prioritization is 16% compared to 6% without, supporting that prioritization achieves higher success rate. An agent should be capable of auditing its own reasoning and plan to ensure that its prioritization correctly targets asynchronous subtasks. We discuss methods for reliable self-verification in Section 5.

**Finding 7.** Stronger priors improve asynchronous performance.

The dominant failures of `gpt4-o ReAct` on the asynchronous dataset were transition failures. We investigate how we can improve performance by increasing the priors over the transition function. We create an augmented method, `ReAct + Prior`, that prompts `ReAct` with more details about the rules of ROBOTOUILLE.

Fig. 7 shows nested pie charts of the failure modes on Tasks 1 to 3 of the asynchronous dataset from the `gpt4-o ReAct` experiments in Table 3 and from `gpt4-o ReAct + Prior`.

We observe a statistically insignificant change in performance, where the success rate for `gpt4-o ReAct` is $0.30 \pm 0.085$ and `gpt4-o ReAct + Prior` is $0.40 \pm 0.050$. We also observe failures

relating to violating the 'one item at station' rule decrease from $38.1\%$ for `gpt4-o ReAct` (8 failures) to $22.2\%$ for `gpt4-o ReAct + Prior` (4 failures) accounting for a $50\%$ decrease in these transition failures. While this shows that increasing priors over rules decreases transition failures as expected, overall performance did not improve due to other failures that arose. We note that state failures increase from $23.8\%$ for `gpt4-o ReAct` (5 failures) to $38.9\%$ for `gpt4-o ReAct + Prior` (7 failures). These failures are due to misunderstandings with the state description provided; specifically, the agent assumes that meat on a stove always implies it is cooked. Augmenting `ReAct + Prior` over state priors would presumably improve performance but is impractical because it requires excessive effort from a domain-expert and wouldn't generalize to new domains. We discuss methods for incorporating state feedback in Section 5.

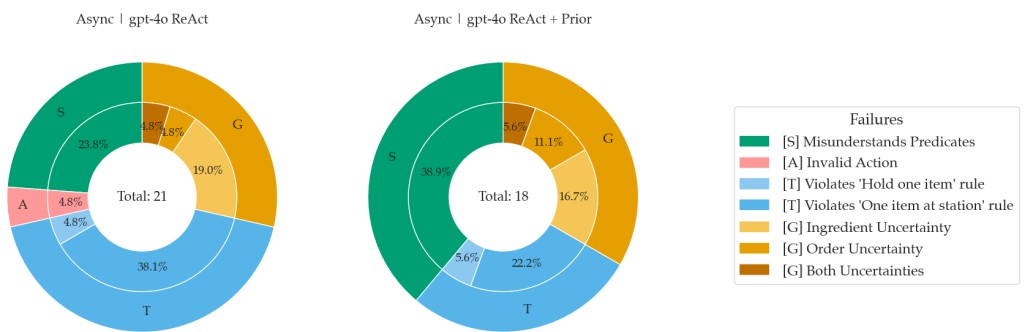

Figure 7: Nested pie chart of failure modes capturing uncertainties in the MDP of `gpt4-o ReAct + Prior` on Tasks 1 to 3 (30 problems) of the asynchronous dataset using `gpt4-o ReAct` and `gpt4-o ReAct + Prior`.

## 5 DISCUSSION

In this paper we propose a new benchmark, ROBOTOUILLE, for stress testing LLM agents on synchronous, asynchronous, and multi-agent settings. We evaluate state-of-the-art LLMs and expose their dominant failure modes are similar across synchronous and asynchronous settings. We perform follow-up studies to bring up performance and uncover the need for improvements in LLM agents that we discuss below.

**Feedback Incorporation** A general method to incorporate long-horizon planning feedback in LLM agents is to include all interactions in the context history. This works well for models with large context windows or near-infinite attention mechanisms (Liu et al., 2023b; Munkhdalai et al., 2024), but LLMs often struggle with long-contexts (Liu et al., 2023c). An alternative is RAG (Lewis et al., 2021), yet this shifts the complexity to retrieval. As explored in Section 4.2.4, a promising approach is for the agent to summarize interactions into facts to reduce uncertainty and strengthen priors. It should also reason about future states to avoid myopic behaviors, as shown qualitatively in Appendix A.11. Another underexplored yet effective approach is finetuning LLM agents (Chen et al., 2023) with methods such as TD learning and value propagation (Putta et al., 2024; Gehring et al., 2024).

**Self-Verification** An LLM agent should be able to audit but LLMs are unreliable at self-verification (Valmeekam et al., 2023a). Other approaches use LLMs to create a representation for external planners (Liu et al., 2023a; Guan et al., 2023) or finetune on planning datasets (Pallagani et al., 2022; Lehnert et al., 2024) but these methods are difficult to debug and lack guarantees respectively. One approach is to combine code-use with language (Wang et al., 2024); reasoning in language and verifying understanding with code and APIs would allow us stronger guarantees that are easier to debug.

**Real-World Application** To effectively deploy LLM agents on real-world agents, the cost and inference time of LLMs must be brought down to make them affordable and quick. This is especially problematic for long-horizon task planning since cost and inference time increases as context grows. These system must also be evaluated with real humans; one future direction for Robotouille is serving as an online platform to test agents with humans through collaboration.

ACKNOWLEDGEMENTS

This work was supported in part by the National Science Foundation FRR (#2327973) and ONR Young Investigator Award. Sanjiban Choudhury is supported in part by the Google Faculty Research Award and the OpenAI Superalignment Grant.

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

# A  APPENDIX

## A.1  RELATED WORKS

In this section we will focus on our desiderata for LLM assistants and how ROBOTOUILLE is different from other related works (Table 1).

**Asynchronous Planning** Many benchmarks evaluate the task planning abilities of LLM agents (Shridhar et al., 2021; Gong et al., 2023; Liu et al., 2018; Valmeekam et al., 2023b; Yao et al., 2024; Zhou et al., 2024; Yao et al., 2023a) but few test the ability to plan asynchronously. Existing work relevant to asynchronous planning evaluate LLM capabilities on temporal logic (Wang & Zhao, 2024) or use graph-based techniques (Wu et al., 2024); (Besta et al., 2024)) but do not focus on it. (Lin et al., 2024) proposes the Plan Like a Graph technique and a benchmark AsyncHow that focuses on asynchronous planning but makes a strong assumption that infinite agents exist. (Carroll et al., 2020) proposes a benchmark, Overcooked-AI, that involves cooking onion soup which has time delays but has limited tasks and focuses on lower-level planning without LLM agents. ROBOTOUILLE has a dataset focused on asynchronous planning that involves actions including cooking, frying, filling a pot with water, and boiling water.

**Diverse Long-Horizon Task Planning** There is vast amount of work that use LLMs to plan (Ahn et al., 2022; Huang et al., 2022; Zeng et al., 2022; Liang et al., 2023; Singh et al., 2022; Song et al., 2023; Yang et al., 2023; Song et al., 2023) but they tend to evaluate on short-horizon tasks with limited diversity in tasks. We present the number of tasks, longest plan horizon, and procedural generation capability of various benchmarks in Table 1 to capture these axes. Notable LLM agent benchmarks that capture these axes include PlanBench (Valmeekam et al., 2023b), WebShop (Yao et al., 2023a), and VirtualHome (Puig et al., 2018). ROBOTOUILLE provides a focused set of diverse long-horizon tasks that can be procedurally generated.

**Multi-agent Planning** LLM agent benchmarks like (Liu et al., 2023d; Xu et al., 2023; Ma et al., 2024; Gong et al., 2023) evaluate multi-agent interactions but do not involve time delays. OvercookedAI (Carroll et al., 2020), while not an LLM agent benchmark, incorporates time delays which brings the complexity of asynchronous planning to multi-agent settings. ROBOTOUILLE provides a multi-agent dataset for 2-4 agents, a choice between turn-based or realtime planning, and incorporates asynchronous tasks for added complexity.

## A.2  ADDITIONAL ROBOTOUILLE JSONS

To provide flexibility in task and environment creation, an environment JSON (Figure 8) is used to define the problem. The size of the grid used can be specified, and positions of objects in the item can be specified using coordinates. Predicates that are specific to an item can also be specified. In conjunction with the flexible goal creation described in Section 2, objects in the environment can be given specific ids, if the goal must be satisfied for specific objects. Additionally, if the environment requires a different number of cuts to complete cutting, or a different cook time, these values can be configured in the JSON.

Adding objects to the environment is also simple. To add a new object, the necessary predicates for that object can be added to the domain JSON, and its corresponding image can be added to the

```
"width": 3,
"height": 3,
"config": {
    "num_cuts": {
        "lettuce": 3,
        "default": 3
    },
    "cook_time": {
        "patty": 3,
        "default": 3
}},
"stations": [{
    "name": "board",
    "x": 0,
    "y": 1,
    "id": "A"
}],
"items": [{
    "name": "lettuce",
    "x": 0,
    "y": 1,
    "stack-level": 0,
    "predicates": ["iscuttable"],
    "id": "a"
}],
"players": [{
    "name": "robot",
    "x": 0,
    "y": 0,
    "direction": [0, 1]
}],
"goal_description": "Cut the lettuce on the board until it is cut",
"goal": [{
    "predicate": "iscut",
    "args": ["lettuce"],
    "ids": ["a"]
}]
```

Figure 8: Environment JSON for a lettuce cutting task.

rendering JSON (Figure 9). If there are different images for the object depending on the predicates that are true in the environment, these can also be specified. The images can also be scaled or offset using the rendering JSON.

To specify what button to press for each action, we use an input JSON (Figure 10). If the action requires a mouse click, we can specify where the player needs to click to perform the action. If the action requires a key press, we specify which button to press for which action, and where the player needs to be to perform the action.

## A.3 ADDITIONAL DATASET DETAILS

**Multiagent Dataset** This dataset consists of tasks designed to test the LLM agent's multiagent capabilities. Robotouille's multiagent capabilities test the agent's ability to collaborate, and is more difficult because it includes tasks where agents may potentially interfere with one another, and share resources with one another.

**Tasks 1 - 3: Burgers** The first 3 tasks involve cooking and assembling a burger with increasing levels of difficulty. In Task 1, the agents need to chop lettuce, and cook a patty, before assembling the burger with the patty, lettuce, a bottom bun, and a top bun. In Task 2, the complexity is increased with an additional ingredient, a tomato, that needs to be cut and stacked onto the burger. In Task 3, lettuce needs to be cut, a chicken needs to be fried, and an onion needs to be cut first before it is fried. This adds a level of complexity because one of the ingredients, the onion, needs to be both cut and fried.

**Tasks 4 - 6: Sandwiches** Tasks 4 - 6 involve making sandwiches. Unlike the tasks which only involve a single agent, sandwiches in multiagent environments are more complex than burgers because there is ambiguity in the stack ordering. In burgers, the bottom bun needs to be at the bottom, while in sandwiches, a piece of bread can be used as either the bottom bread or the top bread. This is complex

```json
"player": {
    "robot": {
        "front": "robot_front.png",
        "back": "robot_back.png",
        "left": "robot_left.png",
        "right": "robot_right.png"
    }},
"floor": "floorkitchen.png",
"item": {
    "constants": {
        "STATION_ITEM_OFFSET" : 0.25,
        "X_SCALE_FACTOR": 0.125,
        "Y_SCALE_FACTOR": 0.75
    },
    "entities": {
        "chicken": {
            "assets": {
                "default": "chicken.png",
                "cooked": {
                    "asset": "cookedchicken.png",
                    "predicates": ["iscooked"]
                },
                "fried": {
                    "asset": "friedchicken.png",
                    "predicates": ["isfried"]
                }
            },
            "constants": {}
        }, ...}},
"station": {
    "constants": {},
    "entities": {
        "fryer": {
            "assets": {
                "default": "fryer.png"
            },
            "constants": {}
        }, ...}}
```

Figure 9: Rendering JSON

```json
"mouse_click_actions": [{
    "name": "move",
    "input_instructions": {
        "button": "left",
        "click_on": "s2"
    }}, ...],
"keyboard_actions": [{
    "name": "cook",
    "input_instructions": {
        "key": "e",
        "at": "s1"
    }}, ...]
```

Figure 10: Input JSON

because the agents need to collaborate and share the resources, and they need to agree on which bread to use as the top bread and the bottom bread. Task 4 involves cutting a lettuce and cooking a chicken before assembling the sandwich. Task 5 involves one more ingredient, a tomato, which also needs to be cut. Task 6 is a double stacked lettuce chicken sandwich. Unlike the previous tasks, Task 6 enforces a strict ordering on the placement of the ingredients, meaning that the agents need to collaborate and properly agree on the order of the ingredients.

**Task 7: Soup** Task 7 involves cooking soup. This involves filling a pot of water, boiling a water, adding three ingredients into the soup: a cut onion, a potato, and a cut tomato, before serving the soup in a bowl. This task is complex because it involves multiple complex actions, including cutting, filling the pot, and boiling the water.

**Tasks 8 - 10: Multiple Recipes** Tasks 8 - 10 involve the agents making multiple recipes. Task 8 involves 2 different recipes, a tomato cheese patty burger, and a onion chicken sandwich. This task is difficult for multiple agents because the 2 recipes both require an ingredient to be cut and an ingredient to be cooked. Task 9 involves making 2 identical lettuce cheeseburgers. Having 2 identical recipes is more complex than having 2 different recipes because the agents need to agree on which ingredient should be used in which burger. Finally, Task 10 involves making 2 different soups, a potato tomato chicken soup, and a potato cheese onion soup. Soups are the most complex recipes because it involves multiple complex actions. Furthermore, once an ingredient has been added to a soup, it cannot be removed. This forces the agents to properly plan for which ingredient should go into which soup.

## A.4 RELATED WORKS TABLE DATA

For each benchmark in (Table 1), we explain how the number of tasks and longest horizon plans were calculated.

### A.4.1 ALFWORLD

ALFWorld consists of 3827 different tasks consisting of 3,553 train tasks, 140 seen tasks, and 134 unseen tasks from the ALFRED dataset.

The longest horizon plan is 50 steps since 50 is the max number of steps per episode in ALFWorld.

### A.4.2 CUISINEWORLD

CuisineWorld consist of 33 unique dishes which represent the tasks.

The longest horizon plan is 11 steps since Figure 2 of CuisineWorld indicates the dish distribution over the number of steps.

### A.4.3 MINIWOB++

MiniWoB++ consist of 40 tasks since 40 tasks are filtered out of 80 total tasks from the MiniWoB benchmark.

The longest horizon plan is 13 steps since Table 1 indicates that 13 is the maximum number of steps needed for a perfect policy to complete the task.

### A.4.4 OVERCOOKED-AI

Overcooked-AI consists of 1 task since onion soup is the only dish in the environment.

The longest horizon plan is 100 steps since 100 is the max number of timesteps that planning methods are evaluated on.

### A.4.5 PLANBENCH

PlanBench consist of 885 tasks consisting of 600 tasks from Blocksworld domain and 285 tasks from the Logistics domain.

The longest horizon plan is 48 steps since Figure 3 in PlanBench indicates that 48 is the longest optimal plan length from both the Blocksworld and Logistics problem sets.

### A.4.6 $\tau$-BENCH

$\tau$-bench consist of 165 tasks consisting of 115 tasks from the $\tau$-retail benchmark and 50 tasks from the $\tau$-airline benchmark.

The longest horizon plan is 30 steps since 30 is the max number of actions per task in $\tau$-bench.

### A.4.7 WEBARENA

WebArena consist of 812 long-horizon web-based tasks.

The longest horizon plan is 30 steps since 30 is the max number of state transitions in WebArena.

### A.4.8 WEBSHOP

WebShop consist of 12087 crowd-sourced text instructions which represent tasks.

The longest horizon plan is 90 steps since 90 is the max number of state visited in Table 2 of WebShop.

### A.4.9 AGENTBENCH

AgentBench consist of 8 environments which represent tasks.

The longest horizon plan is 35 steps since 35 is the largest number of average turns according to table 3 in AgentBench.

### A.4.10 ARA

ARA consists if 12 real-world tasks.

The longest horizon plan is 4 steps after counting the number of steps in the description of each task in Table 1 of ARA.

### A.4.11 ASYNCHOW

AsyncHow consists of 1600 high-quality instances for real-life tasks.

The longest horizon plan is 9+ steps after checking Figure 5 of AsyncHow.

## A.5 MAGIC

MAgIC consists of 5 games which represent tasks.

We will assume all games will have 3 players and the same number of rounds as indicated in Table 3 of magic (1 round for Chameleon, 2 for Undercover, and 5 for Cost Sharing, Prisoner's Dilemma, and Public Good).

Calculations of longest plan with regards to steps:

Chameleon: (3 clues given out to participants + 3 accusations/votes from participants + 1 guess for the final word if the chameleon is correctly identified) * 1 round = 7 steps

Undercover: (3 people are assigned groups + 3 clues are given from participants + 3 votes from participants) * 2 rounds = 18 steps

Cost Sharing: 3 parties get allocation of money + (1 negotiation phase + 1 fairness check) * 5 rounds = 13 steps

Prisoner's Dilemma: 3 decisions from participants * 5 rounds = 15 steps

Public Good: (3 decisions from participants + 1 redistribution of money) * 5 rounds = 20 steps

Therefore, Public Good has the longest horizon plan with 20 steps.

### A.5.1 T-EVAL

T-Eval consists of 23305 tasks according to Table 2 in T-Eval.

The longest horizon plan is 19 steps based on Figure 5b in T-Eval.

### A.5.2 MLAGENTBENCH

MLAgentBench consists of 13 ML tasks from diverse domains ranging in difficulty and recency.

The longest horizon plan is 50 steps based on Figure 7 in MLAgentBench which describes the distribution of numbers of steps used by agents.

### A.5.3 GAIA

GAIA consists of 466 carefully crafted and human annotated questions.

The longest horizon plan is around 45 steps based on Figure 3 in GAIA which describes the distribution of numbers of steps taken and tools used to answer the 466 questions.

### A.5.4 VIRTUALHOME

VirtualHome consists of 2821 programs which represent tasks.

The longest horizon plan is 96 steps after examining all the activities in VirtualHome's Activity Knowledge base and finding the longest.

### A.6 TASK DEPENDENCY GRAPHS

In general, the ordering of ingredients for task dependency graphs does not matter unless specified. For soups, though the task dependency graphs imply a certain order, vegetables can be added to the pot as long as the pot contains water. In addition, all items are placed on the table.

### A.6.1 SYNCHRONOUS GRAPHS

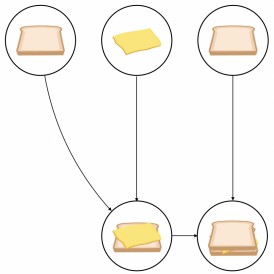

Figure 11: Task 1 for the synchronous dataset in Table 3. The language goal for this graph is "Prepare a cheese sandwich on a table."

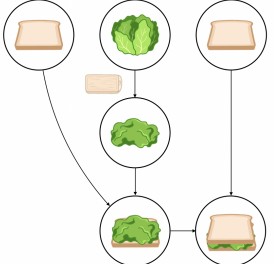

Figure 12: Task 2 for the synchronous dataset in Table 3. The language goal for this graph is "Prepare a lettuce sandwich on a table."

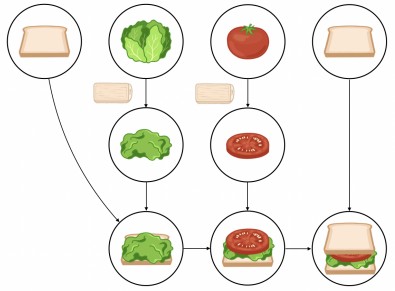

Figure 13: Task 3 for the synchronous dataset in Table 3. The language goal for this graph is "Prepare a sandwich with lettuce and tomato on a table."

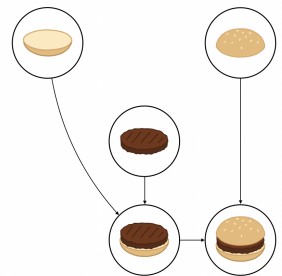

Figure 14: Task 4 for the synchronous dataset in Table 3. The language goal for this graph is "Prepare a hamburger on a table."

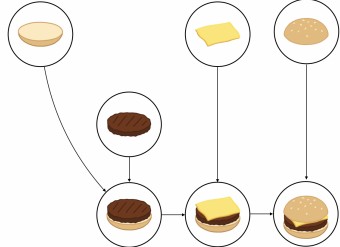

Figure 15: Task 5 for the synchronous dataset in Table 3. The language goal for this graph is "Prepare a cheeseburger on a table."

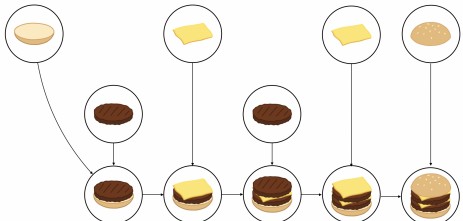

Figure 16: Task 6 for the synchronous dataset in Table 3. The language goal for this graph is "Prepare a double cheeseburger on a table which contains two patties and two cheese slices interleaved (starting with a patty)." This graph also contains the constraint that it needs to be in this exact order.

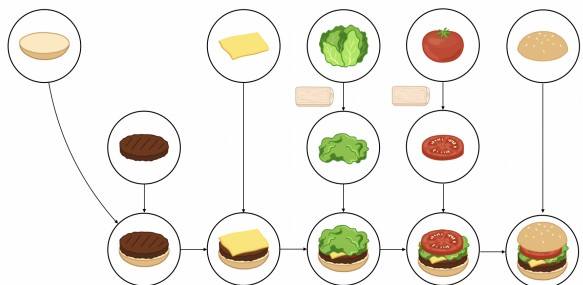

Figure 17: Task 7 for the synchronous dataset in Table 3. The language goal for this graph is "Prepare a lettuce tomato cheeseburger on a table."

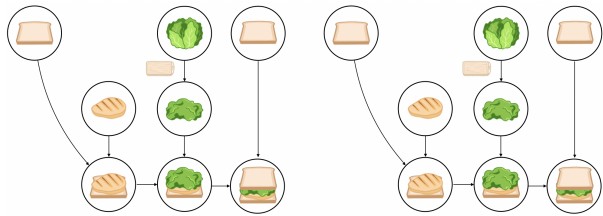

Figure 18: Task 8 for the synchronous dataset in Table 3. The language goal for this graph is "Prepare two lettuce chicken sandwiches on separate tables."

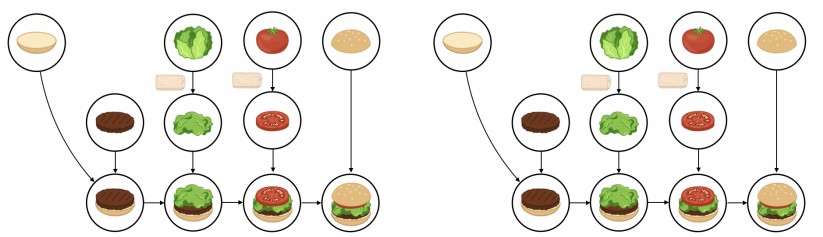

Figure 19: Task 9 for the synchronous dataset in Table 3. The language goal for this graph is "Prepare two lettuce tomato burgers on separate tables."

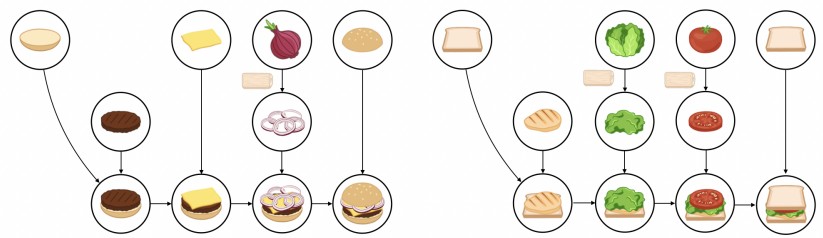

Figure 20: Task 10 for the synchronous dataset in Table 3. The language goal for this graph is "Prepare a burger with cheese and onions on one table and a chicken sandwich with lettuce and tomato on another table."

### A.6.2 ASYNCHRONOUS GRAPHS

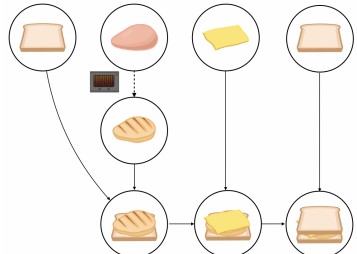

Figure 21: Task 1 for the asynchronous dataset in Table 3. The language goal for this graph is "Prepare a cheese chicken sandwich on a table."

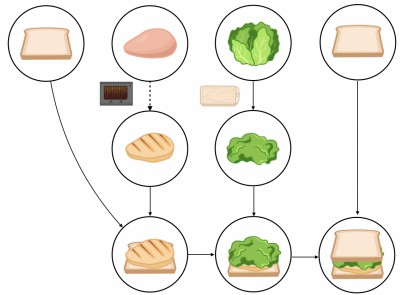

Figure 22: Task 2 for the asynchronous dataset in Table 3. The language goal for this graph is "Prepare a lettuce chicken sandwich on a table."

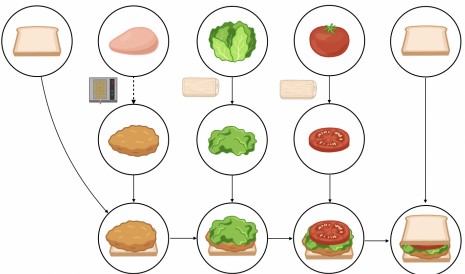

Figure 23: Task 3 for the asynchronous dataset in Table 3. The language goal for this graph is "Prepare a fried chicken sandwich with lettuce and tomato on a table".

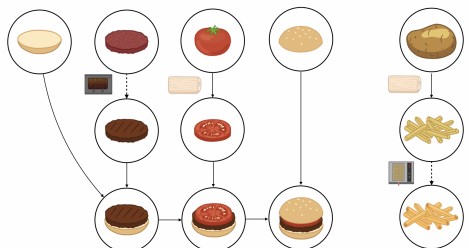

Figure 24: Task 4 for the asynchronous dataset in Table 3. The language goal for this graph is "Prepare a tomato burger and fries on separate tables."

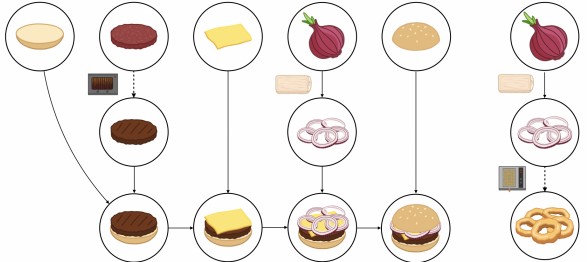

Figure 25: Task 5 for the asynchronous dataset in Table 3. The language goal for this graph is "Prepare an onion cheese burger and fried onion rings on separate tables."

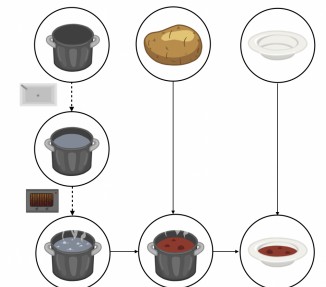

Figure 26: Task 6 for the asynchronous dataset in Table 3. The language goal for this graph is "Make potato soup with a whole potato and serve into a bowl on a table."

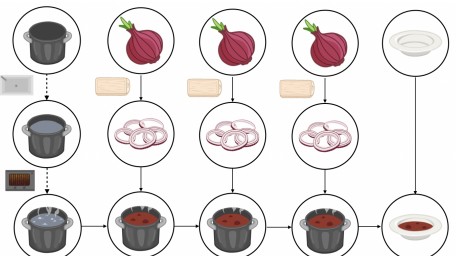

Figure 27: Task 7 for the asynchronous dataset in Table 3. The language goal for this graph is "Make onion soup with 3 cut onions and serve into a bowl on a table."

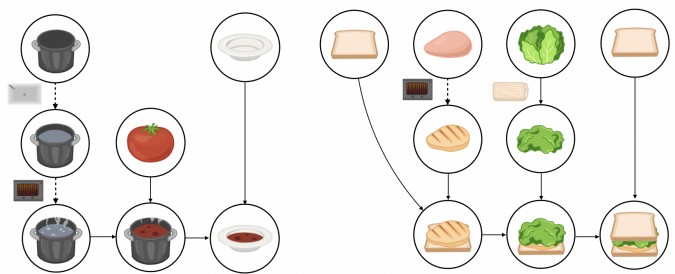

Figure 28: Task 8 for the asynchronous dataset in Table 3. The language goal for this graph is "Make tomato soup with a whole tomato served into a bowl on a table and a lettuce chicken sandwich on another table."

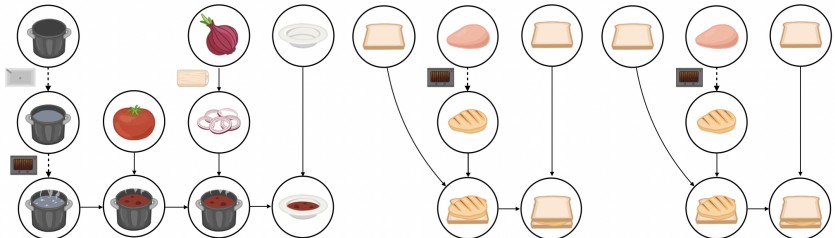

Figure 29: Task 9 for the asynchronous dataset in Table 3. The language goal for this graph is "Make soup with a cut tomato and cut onion served into a bowl on a table and two chicken sandwiches on other tables."

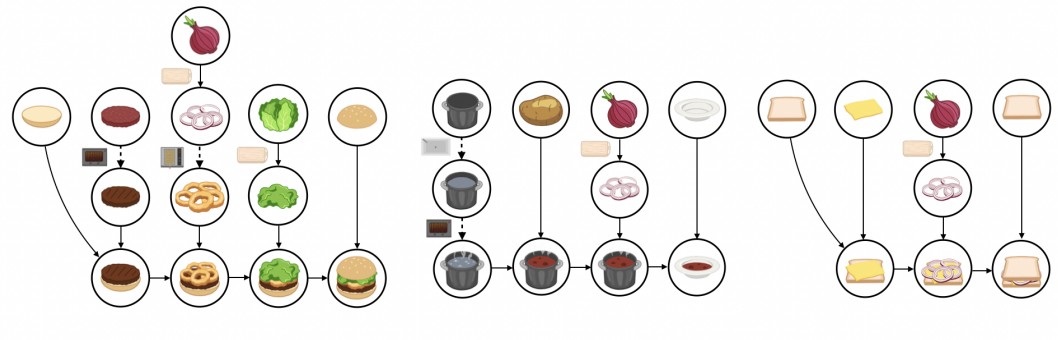

Figure 30: Task 10 for the asynchronous dataset in Table 3. The language goal for this graph is "Make soup with a whole onion and potato served into a bowl, a burger with lettuce and fried onion rings, and an onion chicken sandwich all on separate tables."

### A.6.3 MULTI-AGENT GRAPHS

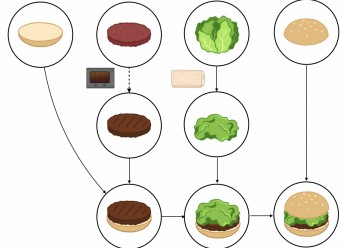

Figure 31: Task 1 for the multi-agent dataset. The language goal for this graph is "Prepare a lettuce burger on a table."

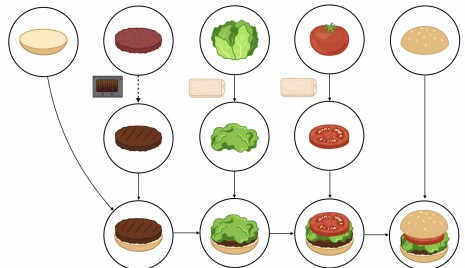

Figure 32: Task 2 for the multi-agent dataset. The language goal for this graph is "Prepare a lettuce tomato burger on a table."

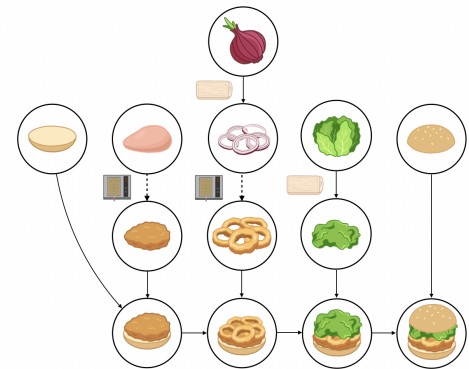

Figure 33: Task 3 for the multi-agent dataset. The language goal for this graph is "Prepare a lettuce fried onion fried chicken burger on a table."

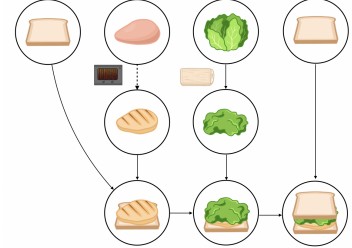

Figure 34: Task 4 for the multi-agent dataset. The language goal for this graph is "Prepare a lettuce chicken sandwich on a table."

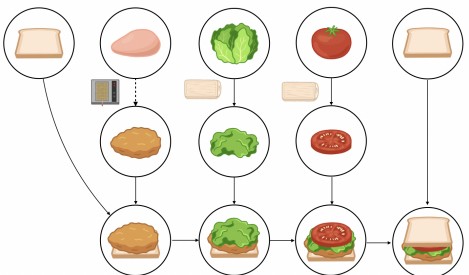

Figure 35: Task 5 for the multi-agent dataset. The language goal for this graph is "Prepare a lettuce tomato fried chicken sandwich on a table."

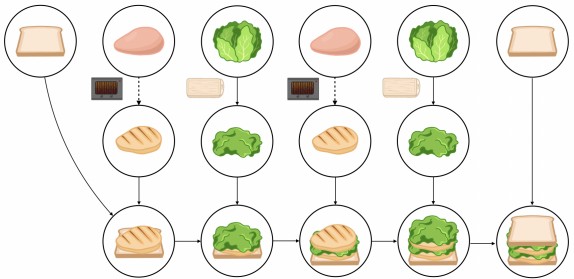

Figure 36: Task 6 for the multi-agent dataset. The language goal for this graph is "Prepare a double lettuce chicken sandwich on a table which contains two chicken patties and two lettuce leaves interleaved (starting with a patty)." This graph also contains the constraint that it needs to be in this exact order.

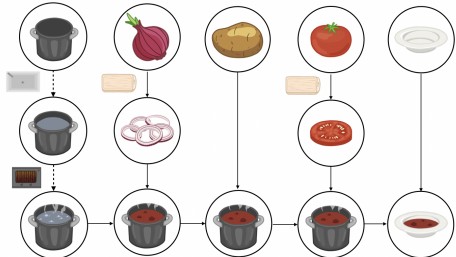

Figure 37: Task 7 for the multi-agent dataset. The language goal for this graph is "Prepare a onion potato tomato soup on a table."

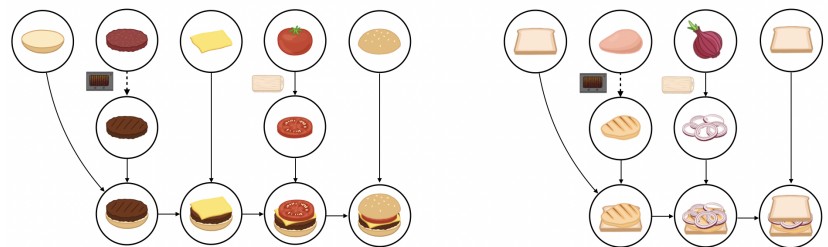

Figure 38: Task 8 for the multi-agent dataset. The language goal for this graph is "Prepare a tomato cheeseburger on one table and a onion chicken sandwich on another table."

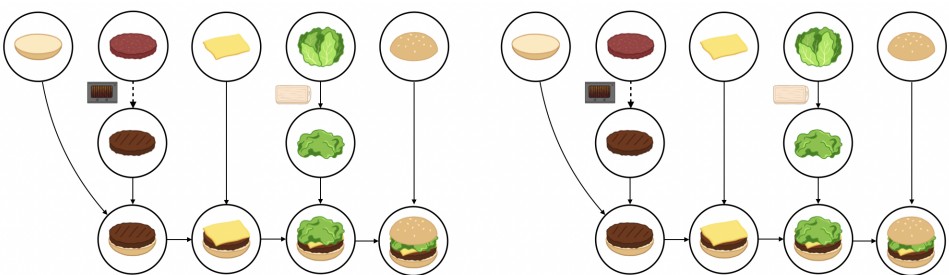

Figure 39: Task 9 for the multi-agent dataset. The language goal for this graph is "Prepare two lettuce cheeseburgers on separate tables."

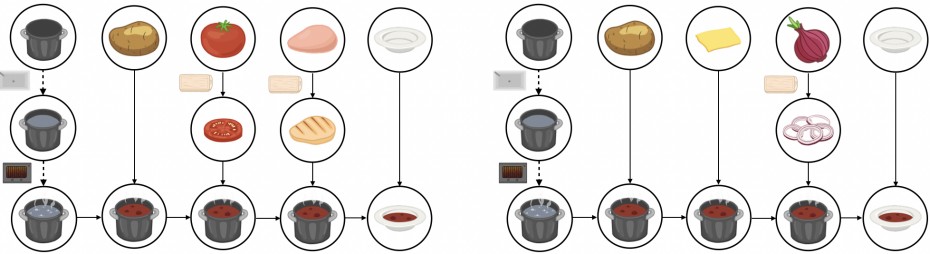

Figure 40: Task 10 for the multi-agent dataset. The language goal for this graph is "Prepare a soup with potato, tomato, and chicken on one table and a soup with potato, cheese, and onion on another table."

## A.7 ReAct Ablations

ReAct in its original form can grow very expensive in cost on long horizon tasks due to the increasing context size. We sought to perform early ablations of ReAct to find a cost-effective variant whose performance is relatively the same. We first ablated on the types of feedback from feedback at all ("no-history") to ablating away components of the feedback from the last time step (where "last-obs-reasoning-action" represents the last timestep with all feedback, "last-reasoning-action" represents the last timestep with only the reasoning and action, and "last-action" represents the last timestep with only the last action. Next, we tested two different types of reasoning; one where we simply prompt ReAct to reason about the given information and another where we make it provide a plan in its sequence before outputting a single action (which we've termed "mpc" after Model Predictive Control). From these ablations on a small subset of data, we determined that "last-reasoning-action-mpc" was the best performing and inexpensive as shown in Table 4.

| Experiment | Accuracy (Tasks Solved) | Average Steps | Cost ($) |
|---|---|---|---|
| no-history | 1/9 | 29.11 | 2.11 |
| no-history-mpc | 1/9 | 28.66 | 2.70 |
| last-action | 1/9 | 29.55 | 2.35 |
| last-action-mpc | 2/9 | 26.66 | 2.92 |
| last-reasoning-action | 1/9 | 28.88 | 2.46 |
| **last-reasoning-action-mpc** | **3/9** | **28.33** | **2.64** |
| last-obs-reasoning-action | 2/9 | 26.77 | 3.02 |
| last-obs-reasoning-action-mpc | 3/9 | 23.88 | 2.93 |

Table 4: Performance comparison across different ablations of ReAct. The variant using "last-reasoning-action" and "mpc" ties for best performance but outperforms others in terms of cost.

## A.8 Differences in Horizon Performance

In Table 3 we observe that horizon length does not necessarily correlate with success. The main confounding variable is the quality of few-shot examples. Each dataset provides a single optimal few-shot example from a training task excluded from the testing set. This example is insufficient when the LLM agent makes a mistake because it has not seen examples of incorporating state feedback to recover from failure. The LLM agent, therefore, acts in an open-loop manner.

In the synchronous dataset, Task 5 is more complex than Task 4, yet it has a higher success rate. This is because Task 5 is more aligned to the few-shot example, sharing a common sub-trajectory (i.e. stacking cheese). This similarity allows ReAct to stay within the distribution of the example, leading to fewer mistakes. In contrast, Task 4 deviates more from the example, resulting in ReAct making mistakes it cannot recover from.

Similarly, in the asynchronous dataset, we also observe that Task 1 < Task 2 < Task 3 despite having increasing complexity. Task 2 and 3 are more aligned to the few-shot example, sharing common

sub-trajectories (i.e. cutting veggies) so we expect the two to perform at least as well as Task 1. We also expect some variance since we run our models with a temperature of 0.7; Tasks 2 and 3 are within standard error (30.0 ± 13.8 for Task 2 versus 40.0 ± 14.8 for Task 3) so they perform similarly.

### A.9  WHY IS ASYNCHRONOUS HARDER THAN SYNCHRONOUS?

The complexity of search for synchronous and asynchronous given the MDP in Section 2 is:

1. Synchronous Case ($d = 0$): No delays, so the planner operates in $O(|S| + |A|)$
2. Asynchronous Case ($d > 0$): Each delay expands the effective state space, yielding $O(|S| \times (d + 1)^n + |A|)$ complexity, where $n$ is the number of timers

Hence the expanded state space requires both a conventional planner or a LLM based planner to reason over a larger range of delayed effects.

### A.10  FAILURE MODE TAXONOMY DETAILS

**Markov Decision Process** To categorise the failure modes, we used the Markov Decision Process (MDP), where there are 4 main failure modes:

1. State failures
2. Action failures
3. Transition Function failures
4. Goal failures

We chose to use the MDP because LLMs know the MDP of the real world very well, but struggle to learn and understand the MDP of new enviornments, such as Robotouille. In using the MDP as a system to categorise failures, we are better able to see how the LLM bridges its knowledge of the real world to its understanding of new environments.

**State Failures** A failure is categorised as a state failure when the agent misunderstands predicates in the state, and fails as a result of this misunderstanding. For example, when preparing a chicken cheese sandwich, the LLM agent may mistake the chicken to be already cooked, when it is not. Even though the predicates are true in the state, the agent misinterprets the predicates, causing it to take the wrong action and preventing it from achieving the goal successfully.

```
Item chicken2:
chicken2 is a chicken breast
chicken2 can be cooked on a stove
chicken2 is at stove4
chicken2 is directly on top of stove4
chicken2 has nothing above it

...

Reasoning: The goal is to make an onion chicken sandwich on a
    table. I can cook chicken while doing other tasks. Thus, I
    will stack the ingredients from bottom to top as follows:
    table -> bottom bun -> chicken -> cheese -> bread.

1. First, I will leave chicken1 cooking. chicken1 is already
    cooked and in the sandwich.
2. The first ingredient is the bottom bun. I see bottombun1 on
    table3 with chicken1 on top of it. I will leave bottombun1
    in place since it is already in the correct position.
3. The second ingredient is chicken. I see chicken1 stacked on
    bottombun1. Since chicken1 is already stacked on
    bottombun1, I will leave it in place.
```

```
4. The third ingredient is cheese. I see cheese1 on table3.
   Since cheese1 is directly on top of chicken1, I will leave
   it in place.
5. The fourth ingredient is bread. I see bread2 on table3 and
   bread1 held by robot1. I need to stack bread1 on top of
   cheese1 to complete the sandwich.

Action: Stack bread1 on top of cheese1 using robot1
```

Listing 1: Example of a State failure: 0_cheese_ chicken_ sandwich_ 252

**Action failures** A failure is categorised by an action failure when the agent exceeds the number of steps and fails to complete the environment because it repeatedly tries to take an invalid action. Even thought the valid actions are listed to the agent at each step, the agent may ignore the prompt and try to take another invalid action repeatedly. For example, when making potato soup, the agent tries to "transfer" the soup in the pot into a bowl it is currently holding, although the correct action in the environment is to pick up the pot and "pour" the soup into the bowl.

```
Valid Actions:
Move robot1 from stove1 to sink1
Move robot1 from stove1 to table1
Move robot1 from stove1 to table2
Move robot1 from stove1 to fryer1
Move robot1 from stove1 to board1
Move robot1 from stove1 to fryer2
Move robot1 from stove1 to stove2
Move robot1 from stove1 to table3
Do nothing

Goal: Make potato soup with a whole potato and serve into a
    bowl on a table.

ACTION PROPOSAL RESPONSE
--------------------

Reasoning: The goal is to make potato soup with a whole potato
    and serve it into a bowl on a table. The steps required to
    achieve this goal are:

1. Fill pot1 with water (done).
2. Boil the water in pot1 (done).
3. Add potato1 to the boiling water in pot1 (done).
4. Serve the soup into bowl1 on a table.

Currently, potato1 has been added to boiling water in pot1,
    and pot1 is at stove1. I am holding bowl1 which is empty.
    The next step is to transfer the soup from pot1 to bowl1.
    Since I am already at stove1, I will transfer the soup from
     pot1 to bowl1.

Action: Transfer water1 and potato1 from pot1 to bowl1 using
    robot1

ACTION PROPOSAL PROMPT
--------------------

Error Feedback: The action 'Transfer water1 and potato1 from
    pot1 to bowl1 using robot1' is not valid. Please provide a
    valid action.
```

Listing 2: Example of an Action failure: 5_ potato_ soup_ 126

**Transition Function failure** A failure is categorised as a transition function failure when the agent violates a rule, even though the rules are included in the prompt to the agent. When the agent tries to plan for and perform actions that break the rule, if it is unable to recover, it will repeat transitions, causing it to exceed the number of steps and fail to complete the task in the environment. For example, one of the rules in the environment is that "A Station must contain a single Item to perform an action on it". When trying to prepare an onion soup, the agent may try to cut multiple onions together on a single cutting board at the same time. While this is possible in the real world, it is not allowed in our environment. This misunderstanding of the transition failure causes the agent to run out of steps.

```
Reasoning: The goal is to make onion soup with 3 cut onions
    and serve it into a bowl on a table. The steps to achieve
    this goal are as follows:

1. Gather and cut the onions.
2. Place the cut onions into a pot with water.
3. Boil the pot with the onions.
4. Serve the soup into a bowl on a table.

Currently, I am holding onion1 and need to cut it. Since I am
    at board1, I will stack onion1 on top of onion2 to cut it
    on the cutting board.

Action: Stack onion1 on top of onion2 using robot1
```

Listing 3: Example of an Transition Function failure: 6_ onion_ soup_ 126

**Goal failure** A failure is categorised as a goal failure when the agent fails due to a misunderstanding of the goal, even though the goal is specified to the agent. This may mean that the agent stacks the ingredients in the wrong order, fail to prepare the ingredients before assembling the recipe, or include ingredients not included in the recipe. For example, when preparing a cheese chicken sandwich, the agent may fail to cook the chicken before assembling the sandwich.

```
Reasoning: The goal is to make a cheese chicken sandwich on a
    table. The ingredients need to be stacked from bottom to
    top as follows: table -> bread -> chicken -> cheese ->
    bread.

Given the current state:
1. I am holding chicken1 at table3.
2. The base of the sandwich includes bread2 and bottombun1 and
     is already correctly stacked on table3.
3. The next step is to stack the chicken on top of the
   existing bottom bun and bread combination.
4. After the chicken, I will need to stack cheese, which is on
     stove3.
5. The final ingredient is the top bun, which is not
    explicitly mentioned but can be assumed to be the topmost
    bread slice.

Therefore, the immediate action will be to stack chicken1 on
    top of bottombun1, which is already at table3.
```

Listing 4: Example of an Goal failure: 0_ cheese_ chicken_ sandwich_ 252

**Prioritising Failure Modes** Sometimes, the agent may fail due to multiple reasons. For example, it may violate rules and also misunderstand the goal. In such cases, we choose the failure

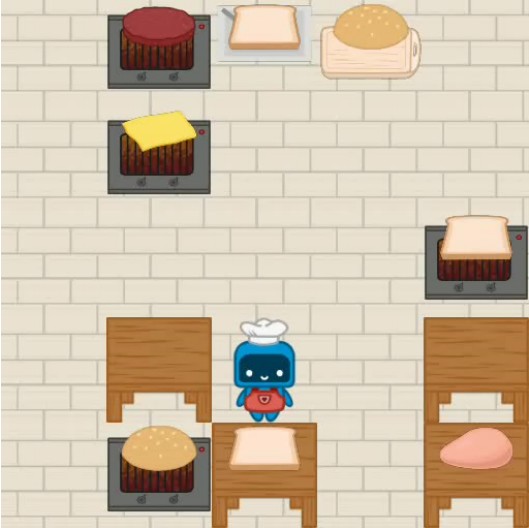

Figure 41: 0_ cheese_ chicken_ sandwich_ 84: All stoves are occupied, and agent naively follows rules, causing it to repeat transitions

mode that ultimately contributes to the goal. If it ultimately tries to perform actions based on a misunderstanding of the state, causing it to fail, we classify it as a state failure. If it repeats transitions due to a misunderstanding of the valid actions in the state, we classify it as an action failure.

The dominant failure modes in the datasets are transition function failures and goal failures, and there are multiple environments where the agent seemingly fails to understand both the transition function and the goal. In this case, we refer to the following tiebreaker. If the agent misunderstands the goal but also violates rules, we find the reason why the agent failed. In the case it violates a rule and is unable to recover, we classify it as a transition function failure because it was unable to ultimately make any progress towards the goal, whether it was accurate or not. If the agent was able to recover but failed due to a misunderstanding of the goal, we classify it as a goal failure.

Another situation that is unclear is when the agent fails to prepare the ingredients before assembling the recipe. This could either be due to a misunderstanding of the state, or because of a misunderstanding of the goal. If the agent reasons that the ingredients is already prepared when it is not, it is classified as a state failure. However, if the agent does not reason about preparing the ingredient at all, then it is classified as a goal failure.

### A.11 QUALITATIVE PLANNING FAILURE EXAMPLE

To gain more insight into why LLM agents are unable to efficiently complete asynchronous tasks successfully, we created a new baseline by repeating the rules to the agent before every action. However, this caused the agent to sometimes be even more inefficient, as it naively tries to follow rules without proper planning.

In this example, the agent needs to cook a chicken on a stove, but all the stoves are currently occupied. It first follows the rule "A Station must contain a single Item to perform an action on it" and picks up the item from the stove. Then, it follows the rule "A Player can only hold a single Item at a time". Since it is currently holding the item it does not need, and it wants to cook the chicken, it places the item back onto the stove to free its hands. This causes the agent to repeatedly pick up the item from the stove and place it back onto the stove. (Fig. 41)

```
Reasoning: The goal is to make a cheese chicken sandwich on a
    table. The sandwich will be stacked in the following order:
    table -> bread -> chicken -> cheese -> bread.
```

```
1. The first ingredient, bread1, is already on table2 and will
     serve as the base for the sandwich. I will leave it in
     place.

2. The second ingredient is chicken. I see chicken1 on table3.
     I must cook chicken1 before stacking it on bread1. stove4
     currently has bread3 on it, which is not needed for the
     sandwich. I will pick up bread3 from stove4 to make space
     for cooking chicken1.

Action: Pick up bread3 from stove4 using robot1

...

Reasoning: The goal is to prepare a cheese chicken sandwich on
     a table. To achieve this, I need to stack the ingredients
     from bottom to top as follows: table -> bread -> chicken ->
     cheese -> bread.

1. The first ingredient, bread1, is already on table2 and will
     serve as the base for the sandwich. I will leave it in
     place.

2. The second ingredient is chicken. I see chicken1 on table3.
     I must cook chicken1 before stacking it on bread1. stove4
     is currently empty, so I will place bread3 there to free up
     my hands.

Action: Place bread3 on stove4 using robot1
```

Listing 5: Log for 0_ cheese_ chicken_ sandwich_ 84

### A.12 QUALITATIVE BAD START FAILURES

**5_ double_ cheeseburger_ 42** In this task, the agent misunderstands the goal from its very first line of reasoning. The goal is to make a double cheeseburger on a table, with a bottom bun, cooked patty, cheese, cooked patty, cheese, and a top bun, stacked in that order. However, the agent misunderstands the goal and does not include all of the ingredients, causing it to fail to complete the task.

```
Reasoning: The goal is to make a double cheeseburger on a
     table with the following stack: table -> patty -> cheese ->
     patty -> cheese.

1. The first ingredient is a patty. I see patty1 on board1 and
     patty2 on table2. Since board1 is a cutting board and not
     a table, I will use patty2 on table2 as the base for the
     burger. I will move robot1 to table2 to pick up patty2.

Action: Move robot1 from table1 to table2
```

Listing 6: 5_ double_ cheeseburger_ 42

**9_ onion_ cheese_ burger_ and_ lettuce_ tomato_ chicken_ sandwich_ 168** In this task, the agent correctly understands the ingredients needed for the two recipes and the order in which they should be stacked, but misunderstands the preparation of the ingredients, and does not register that it needs to first cut the onion, lettuce, and tomato. Thus, it does not include these preparations in its reasoning, and ultimately fails to complete the task.

```
Reasoning: The goal has two parts: prepare a burger with
    cheese and onions on one table, and a chicken sandwich with
     lettuce and tomato on another table.

For the burger with cheese and onions:
1. Bottom bun: bottombun1 on table1.
2. Onion: onion1 on table3.
3. Cheese: cheese1 on table2.
4. Patty: patty1 on fryer1.
5. Top bun: topbun1 on board1.

For the chicken sandwich with lettuce and tomato:
1. Bread: bread1 held by robot1 and bread2 on table4.
2. Chicken: chicken1 on table8.
3. Lettuce: lettuce1 on table5 or lettuce2 on table6.
4. Tomato: tomato1 on sink1.

Steps:
1. Move bread1 (held by robot1) to table4 to combine with
    bread2.
2. Stack bread1 on top of bread2 to complete the bread part of
     the chicken sandwich.
3. Move robot1 to table8 to pick up chicken1.
4. Move robot1 to table4 to stack chicken1 on the bread.
5. Move robot1 to table5 or table6 to pick up lettuce1 or
    lettuce2.
6. Move robot1 to table4 to stack lettuce on the chicken.
7. Move robot1 to sink1 to pick up tomato1.
8. Move robot1 to table4 to stack tomato1 on the lettuce.

For the burger:
1. Move robot1 to table2 to pick up cheese1.
2. Move robot1 to table1 to stack cheese1 on bottombun1.
3. Move robot1 to table3 to pick up onion1.
4. Move robot1 to table1 to stack onion1 on cheese1.
5. Move robot1 to fryer1 to pick up patty1.
6. Move robot1 to table1 to stack patty1 on onion1.
7. Move robot1 to board1 to pick up topbun1.
8. Move robot1 to table1 to stack topbun1 on patty1.

Action: Stack bread1 on top of bread2 using robot1
```

Listing 7: 9_ onion_ cheese_ burger_ and_ lettuce_ tomato_ chicken_ sandwich_ 168

## A.13 STOCHASTIC FAILURES

In our experiments we note that most successes have low repeated transitions while failures have high repeated transitions for both the synchronous and asynchronous datasets (Finding 5, Sec 4.2.3). While this implies that LLM agents aren't effective at following environment constraints, it also reveals a more general issue that LLMs cannot effectively incorporate feedback while generating text. We showcase this by introducing stochasticity into Robotouille; with a 33% chance, a cut item will revert to begin uncut a single time per environment seed. Table 5 shows the results on synchronous tasks that involve cutting for `gpt4-o ReAct` on our original deterministic setting and the stochastic setting. There is a large drop in performance and we have qualitatively found that successes occur when the stochastic change occur immediately, allowing the agent to continue cutting.

| | Deterministic | Stochastic |
|---|---|---|
| **Synchronous (%)** | | |
| [2] 🥪(🥬) | 80.0 | 10.0 |
| [3] 🥪(🥬🍅) | 80.0 | 20.0 |
| [7] 🍔(🥩🧀🥬🍅) | 50.0 | 0.00 |
| [8] 🍔(🥪🥬🥪🥬) | 30.0 | 10.0 |
| [9] 🍔(🥩🥬🍅🍔🥩🥬🍅) | 20.0 | 10.0 |
| [10] 🍔(🥩🧀🧅🥪🥪🥬🍅) | 20.0 | 0.00 |
| **Total** | 56.00 | 1.00 |

Table 5: `gpt4-o ReAct` performance on a subset of synchronous tasks with stochasticity.

## A.14 TRANSITION FAILURE RECOVERY ANALYSIS

In this section, we annotated for the transition failures on the synchronous and asynchronous datasets whether (1) the LLM agent recovers from a failure and (2) whether it repeats its mistake after recovering.

A mistake occurs when the agent violates a rule at a certain station for a specific action. When the agent makes a mistake, there are 4 cases:

1. The agent violates a rule and is unable to recover

2. The agent violates a rule at a station for a specific action, but is able to recover. After recovery, they do not make any more mistakes; they do not repeat the mistake after recovering.

3. The agent violates a rule at a station, recovers, but is later repeats the mistake by trying to perform the same action at the same type of station. In this case, they repeat the mistake after recovering.

4. The agent violates a rule at a station, recovers, and does not repeat the mistake by trying to violate the same rule for the same action at the same type of action. However, they violate the same rule for a different action at a different type of station. In this case, we say that they do not repeat their mistake.

On the synchronous dataset, the transition failures account for 32.1% (17) of the total failures. Of these failures, 58.8% (10) recovered from the mistake. Of the failures that recovered from their mistake, 90% (9) did not repeat the same mistake.

On the asynchronous dataset, the transition failures account for 58.5% (52) of the total failures. Of these failures, 40.4% (21) recovered from the mistake. Of the failures that recovered from their mistake, 57.1% (12) did not repeat the same mistake.

In the case where the agent is able to recover from a mistake, the agent may still fail to complete the task because they recovery process took too long and exhausted the step limit. Then, this failure would be categorised as a Transition Function failure.

## A.15 ADDITIONAL BASELINES FOR ASYNCHRONOUS PLANNING

We attempted to increase performance on the asynchronous dataset by benchmarking 2 new baselines. We introduced PLaG (BaG) Lin et al. (2024), which creates an explicit graph representation of the task with an adjacency list. We adapt this method to multi-turn settings by making `ReAct` construct the graph representation at each step. We also introduce Reflexion Shinn et al. (2023) by allowing `ReAct` 2 attempts at an environment, keeping a reflection of the first failed attempt in context. We present results in Table 6.

| | ReAct | PLaG (BaG) | Reflexion |
|---|---|---|---|
| **Asynchronous (%)** | | | |
| [1] | 20.0 | 20.0 | **30.0** |
| [2] | 30.0 | 30.0 | **50.0** |
| [3] | **40.0** | 20.0 | **40.0** |
| [4] | 10.0 | **30.0** | 10.0 |
| [5] | 0.00 | 10.0 | 10.0 |
| [6] | 10.0 | 20.0 | **30.0** |
| [7] | 0.00 | 0.00 | 0.00 |
| [8] | 0.00 | 0.00 | 0.00 |
| [9] | 0.00 | 0.00 | 0.00 |
| [10] | 0.00 | 0.00 | 0.00 |
| **Total** | 11.0 | 13.0 | **17.0** |

Table 6: `gpt4-o` performance on the asynchronous dataset over various baselines.

We observe that PLaG (BaG) achieves a 2% performance increase over `ReAct` and Reflexion achieves a 6% performance increase over `ReAct`. We expected PLaG (BaG) to perform relatively similar to our `ReAct` baseline because our few-shot examples for `ReAct` construct a language graph representation for the task. Reflexion has a larger performance increase but at the cost of retrying runs which is very expensive for long horizon tasks. These results highlight that future work should investigate how LLM agents can optimally represent asynchronous tasks and cost-effective methods for incorporating feedback to recover from errors.

### A.16 ADDITIONAL RESULTS FROM OPEN-SOURCE MODELS

We perform experiments on synchronous and asynchronous settings for top-performing open-source LLMs on the HuggingFace leaderboard with the ReAct baseline. The following experiments were run on 2 or 4 NVIDIA RTX 6000 Adas using FP8 quantization.

| | Synchronous (%) | Asynchronous (%) |
|---|---|---|
| Qwen2-72B-Instruct | 7.00 | 2.00 |
| Qwen2-32B-Instruct | 6.00 | 1.00 |
| Meta-Llama-3.1-70b-Instruct | 2.00 | 0.00 |
| Meta-Llama-3.1-8b-Instruct | 1.00 | 0.00 |
| gemma-2-27b-it | 0.00 | 0.00 |
| gemma-2-9b-it | 0.00 | 0.00 |

Table 7: Success rates of state-of-the-art open-source LLMs on the synchronous and asynchronous datasets.

### A.17 SYNCHRONOUS VARIANTS OF ASYNCHRONOUS TASKS

We perform a closer analysis of the difficulty between the synchronous and asynchronous datasets by adapting the first 3 tasks of the asynchronous dataset to synchronous variants. We do this by making cooking instant by setting the time delay to 0. The results in Table 8 demonstrate that `ReAct` `gpt4-o` performs better in the synchronous variants compared to the asynchronous setting.

| | Synchronous (%) | Asynchronous (%) |
|---|---|---|
| [1] 🥪(🥖🧀) | 50.0 | 20.0 |
| [2] 🥪(🥖🥦) | 60.0 | 30.0 |
| [3] 🥪(🥖🥦🍅) | 50.0 | 40.0 |

Table 8: ReAct `gpt4-o` performance on 3 synchronous variant tasks for comparison.

