# OpenReview forum: "Robotouille: An Asynchronous Planning Benchmark for LLM Agents"
_ICLR.cc/2025/Conference — ICLR 2025 Poster_

### Official Review · Reviewer_nvj3 · 2024-11-01

**Soundness:** 3
**Presentation:** 4
**Contribution:** 2
**Rating:** 6
**Confidence:** 4

**Summary:**

This paper introduces Robotouille, a benchmark designed to evaluate the planning capabilities of LLMs in asynchronous, long-horizon, and multi-agent environments. Authors highlight a significant performance gap for SOTA approaches (e.g., GPT-4o+ReAct), showing success rates of 47% in synchronous tasks and only 11% in asynchronous ones. The benchmark includes datasets with varying levels of complexity, along with a thorough failure analysis, offering key insights into where and why current approaches fall short.

**Strengths:**

+ Well-written with clear illustrations of domain complexities.
+ Rigorous experimental design, featuring carefully curated datasets across multiple complexity levels and a detailed failure analysis.

**Weaknesses:**

* The paper can be split into two main sections: 1) introducing and baselining the benchmark and 2) analyzing results. While the first part is strong, insights in the second part are somewhat obscured. For instance, in Q6, the majority of both successful (72.7%) and failed (52.8%) trajectories prioritized subtask completion. Does this indicate that subtask prioritization may not be a critical area for improvement in planning? I encourage authors to capitalize more on their analysis.

* The absence of stochastic elements in the environment and experimentation is notable. One of the significant challenges for LLM architectures lie in adapting to stochastic domains [1].

* Although the paper initially emphasizes the multi-agent aspect of the environment, this is not reflected in the empirical results.

**Minor Suggestions**

* Include GAIA benchmark in your related work section (https://arxiv.org/abs/2311.12983).
* Use vectorized image formats to ensure quality remains high when zooming.
* Line 443: Add context to the caption: "The median separates failures into runs with lower and higher repeated transitions, with quartiles representing the mean of these two groups."

[1] David Brandfonbrener, Alberto Bietti, Jacob Buckman, Romain Laroche, and Joan Bruna. "When does return-conditioned supervised learning work for offline reinforcement learning?" In Alice H. Oh, Alekh Agarwal, Danielle Belgrave, and Kyunghyun Cho, editors, Advances in Neural Information Processing Systems, 2022. URL: https://openreview.net/forum?id=XByg4kotW5.

**Questions:**

* Line 366: Could you clarify why (0.5, 1.0] implies minimal progress? Wouldn’t a score of 0.5 indicate the solution is halfway there?

* Line 520: I was surprised not to see value propagation and temporal difference learning listed under Feedback Incorporation techniques.

---

> ### Author Response · Authors · 2024-11-18
>
> Thank you for taking the time to review our work. We will be updating the paper draft with our responses to improve its clarity. We also appreciate your concrete suggestions and have added GAIA to our table, converted our images to SVG, and clarified Figure 6’s caption.
>
> > The paper can be split into two main sections: 1) introducing and baselining the benchmark and 2) analyzing results. While the first part is strong, insights in the second part are somewhat obscured. For instance, in Q6, the majority of both successful (72.7%) and failed (52.8%) trajectories prioritized subtask completion. Does this indicate that subtask prioritization may not be a critical area for improvement in planning? I encourage authors to capitalize more on their analysis.
>
> We have updated the paper with a clarified hypothesis in Q6 and include a clearer metric to support it. In Q6, we investigate how success rate changes with asynchronous task prioritization to understand the impact of asynchronous planning on the results. Our hypothesis is that prioritizing asynchronous subtasks leads to higher success rates as the planned trajectory is shorter and reaches the goal within the maximum step limit. We find that the success rate conditioned on prioritization is 16% compared to 6% without. This more clearly quantifies that task prioritization has a significant impact on success rate.
>
> Please let us know about any other parts of our results analysis that can be improved so we can further strengthen the analysis.
>
> > The absence of stochastic elements in the environment and experimentation is notable. One of the significant challenges for LLM architectures lie in adapting to stochastic domains [1].
>
> We initially avoided introducing stochasticity to minimize confounds and focus solely on investigating the asynchronous planning capabilities of LLMs. It is straightforward to incorporate stochasticity into Robotouille as we demonstrate below on a small subset of synchronous problems.
>
> In the Stochastic Synchronous setting, we introduce a 33% chance that a cut item will revert to being uncut once per environment seed. We hypothesize that the LLM agent will follow its plan in an open-loop manner, without adapting to such changes. In the table below, Tasks 2 and 3 have a statistically significant performance drop. The LLM agent did indeed continue its plan without incorporating the stochastic change.  In Task 2, the agent's only success occurred because it completed the task before the stochastic change happened. In Task 3, the change occurred immediately after an item was cut, allowing the agent a brief opportunity to adapt. We are happy to extend this for all synchronous tasks and add it to the paper.
>
> | Task | Synchronous (%) ± SE | Stochastic Synchronous (%) ± SE |
> |------|----------------------|---------------------------------|
> | 2    | 80 ± 12.0            | 10 ± 9.1                        |
> | 3    | 80 ± 12.0            | 20 ± 12.0                       |
>
> > Although the paper initially emphasizes the multi-agent aspect of the environment, this is not reflected in the empirical results.
>
> Robotouille supports multi-agent environments and provides a dataset of 10 tasks for this setting; however, due to the low performance of LLM agents in the asynchronous setting, we defer this investigation to future work to focus on asynchronous planning. We expect the multi-agent setting to be strictly harder since it involves asynchronous planning.
>
> > Line 366: Could you clarify why (0.5, 1.0] implies minimal progress? Wouldn’t a score of 0.5 indicate the solution is halfway there?
>
> The number indicates normalized “steps to go”, i.e. “in a failed attempt, (optimal distance from terminal state to goal / optimal distance from start to goal)”. Higher is worse, i.e. 0 implies success, >=1 implies that when the agent failed it was further away from the goal than at the start. Hence the (0.5, 1.0] bucket groups failures that are more than halfway at the goal.
>
> We have updated the draft with a clear formula for this metric.
>
> > Line 520: I was surprised not to see value propagation and temporal difference learning listed under Feedback Incorporation techniques.
>
> The Feedback Incorporation section was biased on in-context learning with LLM agents due to the prominence of prompting-based approaches in existing work [1]. However, we acknowledge that fine-tuning LLM agents is a crucial future direction and will add this to our discussion. Training LLM agents with RL techniques like TD learning and value propagation [2,3] remains an important and underexplored area for further research.
>
> [1] FireAct: Toward Language Agent Fine-Tuning. Chen et. al 2023
>
> [2] Agent Q: Advanced Reasoning and Learning for Autonomous AI Agents. Putta et. al 2024
>
> [3] RLEF: Grounding Code LLMs in Execution Feedback with Reinforcement Learning. Gehring et. al 2024.

---

> > ### Author Response · Authors · 2024-11-22
> >
> > We thank the reviewer again for their feedback. Since the discussion period is coming to a close, please let us know if there are any additional questions that we can answer. If you are satisfied with the responses, please consider revising your score to reflect this.

---

> > > ### Comment · Reviewer_nvj3 · 2024-11-27
> > >
> > > Appreciate the detailed responses to my review, and apologies for the delay in my reply. Please find my comments below:
> > >
> > > > We find that the success rate conditioned on prioritization is 16% compared to 6% without. This more clearly quantifies that task prioritization has a significant impact on success rate.
> > >
> > > This analysis is certainly valuable and highlights the importance of prioritization. However, it appears that prioritization alone may not fully explain the observed drop in success rate. Have you considered investigating other potential contributing factors? Broadly speaking, the paper would become even more impactful if you could systematically extract and articulate the key takeaways from each experiment. This would enhance the reader's understanding of the practical insights derived from your findings.
> > >
> > > > Stochasticity MDPs
> > >
> > > Thank you for conducting the new set of results. I would recommend starting with lower noise levels (e.g., 5-10%) to establish a baseline for comparison. Nonetheless, the primary observation should remain consistent: planners are expected to face greater challenges as stochasticity increases. Highlighting this trend with concrete examples could make the results more compelling.
> > >
> > > > Robotouille supports multi-agent environments and provides a dataset of 10 tasks for this setting; however, due to the low performance of LLM agents in the asynchronous setting, we defer this investigation to future work to focus on asynchronous planning. We expect the multi-agent setting to be strictly harder since it involves asynchronous planning.
> > >
> > > Given this deferral, I would suggest toning down the emphasis on the multi-agent aspect throughout the paper. Overemphasizing it might lead readers to expect a stronger baseline or a more detailed analysis than what is currently presented.
> > >
> > > Overall, my main scores remain unchanged. The environment is an excellent testbed for assessing the planning capabilities of LLMs. However, the paper would significantly benefit from stronger and more explicitly articulated takeaways to enhance its utility for readers.

---

> > > > ### Author Response · Authors · 2024-11-28
> > > >
> > > > We appreciate your follow-up feedback and have systematically extracted the key takeaways from our analyses, which we have summarized in the *Overall Results* section of the paper. For your visibility, we include them below:
> > > >
> > > > - **Closed-loop agents are superior**: The best baseline, gpt-4o ReAct, achieves 47% on the synchronous dataset and 11% on the asynchronous dataset, surpassing open-loop approaches IO and IO CoT (Finding 1, Sec 4.2.2).
> > > > - **Poor feedback incorporation leads to decreased asynchronous performance**: Despite being closed-loop, gpt-4o ReAct failures often make little progress towards the goal (Finding 3, Sec 4.2.2) due to poor failure recovery (Finding 5, Sec 4.2.3). We find that boosting priors improves performance (Finding 7, Sec 4.2.4) but discuss better feedback methods in Section 5.
> > > > - **Synchronous and asynchronous failures are closely related**: Both synchronous and asynchronous failures are dominated by rule violations and goal misinterpretation (Finding 4, Sec 4.2.2). We hypothesize that this is due to poor failure recovery (Finding 5, Sec 4.2.3) and agents that recover efficiently could boost performance in both settings.
> > > > - **Task prioritization is critical in asynchronous planning**: Proper prioritization of subtasks in asynchronous settings significantly boosts performance (Finding 6, Sec 4.2.4).
> > > >
> > > >
> > > > We also agree with the suggestion to incorporate the stochastic results into the main paper. These results will strengthen our argument regarding our baseline agent making ineffective use of feedback.

---

### Official Review · Reviewer_ZuBC · 2024-11-04

**Soundness:** 3
**Presentation:** 3
**Contribution:** 3
**Rating:** 8
**Confidence:** 4

**Summary:**

This paper introduces ROBOTOUILLE, a challenging benchmark environment designed to evaluate the capabilities of large language model (LLM) agents in handling asynchronous, long-horizon, and multi-agent scenarios. Specifically, ROBOTOUILLE serves as a simulator for cooking various recipes, aiming to stress-test LLM agents. It utilizes domain and problem JSONs to define the Markov Decision Process (MDP) and the language description of the environment and tasks. By testing models such as gpt-4-o in both synchronous and asynchronous datasets, the authors demonstrate that agents face greater challenges in completing assigned tasks under asynchronous conditions. Additionally, the paper provides a thorough analysis of failure modes towards the end.

**Strengths:**

1. There is a notable gap in benchmarks for planning and decision-making tasks that involve asynchronous, long-horizon scenarios. The authors effectively compare their work to relevant literature, underscoring the significance of ROBOTOUILLE. Moreover, the cooking aspect of the tasks makes them relatable and easy to understand.

2. By conducting experiments across two different settings, the authors identify key reasons for failure modes in the asynchronous decision-making context, such as the ineffectiveness of failure recovery strategies. This provides valuable directions for future research in asynchronous planning.

3. The models used in the experiments represent the latest advancements in the field, showcasing the highest capabilities of current LLMs. Despite this, the agents still struggle to consistently complete asynchronous tasks, highlighting the relevance of the research direction proposed in the paper.

4. The authors supply essential code and detailed implementation explanations, demonstrating that the experimental results are reproducible and persuasive.

**Weaknesses:**

1. The paper lacks a discussion on prompt design. The performance differences between the synchronous and asynchronous datasets may stem from variations in prompts, which often determine the practical limits of an agent's capabilities.

2. Additionally, the tasks in the synchronous and asynchronous datasets differ, making it inappropriate to directly compare results from both settings during the analysis.

3. If possible, I suggest including experimental results from some open-source models in the model list. While these results may not surpass those of proprietary models, they could encourage the open-source community to pay more attention to this research area.

**Questions:**

1. The paper states that ROBOTOUILLE supports multi-agent environments by simply adding more players to the problem JSON. However, I did not see any related attempts or studies mentioned. Is this intended as a future research direction?

2. Some results in the experiments struck me as peculiar. For instance, in Table 2, the Gemini model performs significantly better than ReAct using the I/O CoT method. What accounts for this discrepancy?

3. Furthermore, in Table 3, Task 5 under synchronous conditions is more complex than Task 4, yet ReAct has a higher success rate. A similar situation occurs in the asynchronous setting. The paper does not provide a clear explanation for this phenomenon. What might be the underlying reasons?

---

> ### Author Response · Authors · 2024-11-19
>
> Thank you for your help in improving our paper! We have updated the paper to clarify many of your points as we detail below.
>
> > The paper lacks a discussion on prompt design. The performance differences between the synchronous and asynchronous datasets may stem from variations in prompts, which often determine the practical limits of an agent's capabilities.
>
> This is a great point and we have added this to the paper in Appendix A.3. For both datasets, we use the same set of instructions as input to I/O, I/O CoT, and ReAct as shown in Appendix A.3. Each method is few-shot prompted with an example optimal trajectory on a task unique to the dataset being evaluated on. Notably, the asynchronous example involves a time delay. We ensured that prompt formatting in both datasets was consistent and will include examples in the Appendix to make this transparent.
>
> > Additionally, the tasks in the synchronous and asynchronous datasets differ, making it inappropriate to directly compare results from both settings during the analysis.
>
> We agree that direct comparison of results between the synchronous and asynchronous dataset would be an inappropriate apples to oranges comparison. Our intention when designing our datasets was to ensure that LLM planning capabilities would be stress tested on unique tasks such that an LLM agent performing well on the synchronous and asynchronous datasets could plan effectively in either setting. We have reviewed and edited the paper to remove any implications of such comparison.
>
> We do, however, claim that asynchronous settings are strictly harder than synchronous settings. For some MDP with state space S and action space A, in synchronous settings, a classical planner (i.e. BFS) has a complexity of $O(|S| + |A|)$ whereas in the asynchronous setting with delay d the complexity increases to $O(|S| \times (d+1)^n + |A|)$ where n is the number of delays. This demonstrates the added computational burden in asynchronous scenarios.
>
> To support this empirically, we adapt Tasks 1-3 in the asynchronous dataset to synchronous variants where there are no time delays (i.e. meat is cooked immediately). The results below demonstrate that ReAct gpt-4o performs better in the synchronous variants.
>
> | Task         | Asynchronous (%) ± SE | Synchronous (%) ± SE |
> |--------------|-----------------------|----------------------|
> | Task 1       | 20.0 ± 12.0           | 50.0 ± 15.1          |
> | Task 2       | 30.0 ± 13.8           | 60.0 ± 14.8          |
> | Task 3       | 40.0 ± 14.8           | 50.0 ± 15.1          |
>
>
> > If possible, I suggest including experimental results from some open-source models in the model list. While these results may not surpass those of proprietary models, they could encourage the open-source community to pay more attention to this research area.
>
> We fully agree and intend to incorporate open-source models into the results. We believe fine tuning open-source models for Robotouille is a promising direction for future research. Below are new experiments on the synchronous setting using top-performing open-source LLMs on the HuggingFace LLM leaderboard. The following experiments were run on 4 NVIDIA RTX 6000 Adas using FP8 quantization. We will be continuing these experiments for the remaining synchronous tasks and the entire asynchronous dataset.
>
> | Model                       | **Synchronous** (%) ± SE   |
> |-----------------------------|----------------------------|
> |                             | Task 1                     |
> | gemma-27b–it                | 20.0 ± 12.0                |
> | gemma-9b–it                 | 0.00 ± 0.00                |
> | Llama-3.1-70B-Instruct      | 30.0 ± 13.8                |
> | Llama-3.1-8B-Instruct       | 10.0 ± 9.10                |
> | Qwen2-72B-Instruct          | 60.0 ± 14.8                |
> | Qwen2-32B-Instruct          | 50.0 ± 15.1                |
>
>
> > The paper states that ROBOTOUILLE supports multi-agent environments by simply adding more players to the problem JSON. However, I did not see any related attempts or studies mentioned. Is this intended as a future research direction?
>
> This is intended as a future research direction. We decided to focus on asynchronous planning due to the low performance of LLM agents in this setting and because we expect the multi-agent setting to be strictly harder since it involves asynchronous planning.

---

> > ### Author Response · Authors · 2024-11-19
> >
> > > Some results in the experiments struck me as peculiar. For instance, in Table 2, the Gemini model performs significantly better than ReAct using the I/O CoT method. What accounts for this discrepancy?
> >
> > The Gemini model notably performs poorly with ReAct. We qualitatively observe that Gemini ReAct erroneously plans to solve the goal of the few-shot example rather than the current environment goal. We suspect this is due to the long context length of the ReAct few-shot example compared to the examples for I/O and I/O CoT. This aligns with findings in [1] where LLMs have been observed to perform poorly in long-context tasks.
> >
> > [1] Lost in the Middle: How Language Models Use Long Contexts. Liu et. al 2023.
> >
> > > Furthermore, in Table 3, Task 5 under synchronous conditions is more complex than Task 4, yet ReAct has a higher success rate. A similar situation occurs in the asynchronous setting. The paper does not provide a clear explanation for this phenomenon. What might be the underlying reasons?
> >
> > We have updated the paper with the following discussion. The main confounding variable is the quality of few-shot examples. Each dataset provides a single optimal few-shot example from a training task excluded from the testing set. This example is insufficient when the LLM agent makes a mistake because it has not seen examples of incorporating state feedback to recover from failure. The LLM agent, therefore, acts in an open-loop manner.
> >
> > In the synchronous dataset, Task 5 is more complex than Task 4, yet it has a higher success rate. This is because Task 5 is more aligned to the few-shot example, sharing a common sub-trajectory (i.e. stacking cheese). This similarity allows ReAct to stay within the distribution of the example, leading to fewer mistakes. In contrast, Task 4 deviates more from the example, resulting in ReAct making mistakes it cannot recover from.
> >
> > Similarly, in the asynchronous dataset, we also observe that Task 1 < Task 2 < Task 3 despite having increasing complexity. Task 2 and 3 are more aligned to the few-shot example, sharing common sub-trajectories (i.e. cutting veggies) so we expect the two to perform at least as well as Task 1. We also expect some variance since we run our models with a temperature of 0.7; Tasks 2 and 3 are within standard error (30.0 ± 13.8 for Task 2 versus 40.0 ± 14.8 for Task 3) so they perform similarly.

---

> > > ### Author Response · Authors · 2024-11-22
> > >
> > > Thank you again for your feedback. Since the discussion period is coming to a close, please let us know if there are any additional questions that we can address for you.

---

### Official Review · Reviewer_qY1d · 2024-11-04

**Soundness:** 2
**Presentation:** 3
**Contribution:** 2
**Rating:** 3
**Confidence:** 4

**Summary:**

This paper proposes a benchmark to evaluate LLMs’ planning ability, focusing in particular on their asynchronous planning capabilities. Here asynchronous means that the optimal solution is not to complete sub-tasks in a sequential manner. Rather, while “waiting for” some subtasks to complete, the agent should directly focus on other tasks. The benchmark contains 30 tasks, where existing LLMs suffer to perform well even with techniques such as chain-of-thought.

**Strengths:**

- The paper proposes a benchmark for asynchronous planning based on an environment similar to overcooked.
- The paper demonstrates that many existing LLMs fail short at planning problems that requires to consider time delays of sub-tasks.

**Weaknesses:**

- While this paper proposes a new benchmark for certain types of LLM reasoning tasks, it does not include sufficient experimental evaluation and analysis to highlight its main challenges. Specifically, while there are many LLMs as well as LLM-based planning algorithms to tackle planning problems, the paper only experimented with one LLM and two planning algorithms (CoT and ReAct). More baselines are needed to better illustrate the challenges posed by the proposed benchmark.

- It is unclear from the paper which components of asynchronous planning make it much harder than synchronous planning. In the failure mode analysis, it is unclear to me what does it mean to attribute failure in terms of uncertainty over the MDP. Additionally, this categorization seems to be irrelevant to the key ingredient of the benchmark: *asynchronous* planning. A detailed analysis of why “asynchronous” planning is harder is needed to understand in what ways do this benchmark challenges the methods.

- While the paper highlights that the benchmark requires long-horizon planning, the horizon counted by the paper is the number of required “atomic steps”, while the agent is performing high-level actions such as move an object to a place. This size of the graphs (e.g., in Figure 11) seems to be a better measure of the “planning depth”. Given this, it seems unclear what are the additionally challenges in this benchmarks compared to for example the AsyncHow dataset mentioned in the paper.

**Questions:**

Which components of asynchronous planning make it much harder than synchronous planning?

---

> ### Author Response · Authors · 2024-11-15
>
> Thank you very much for your thoughtful review. We are grateful for your detailed feedback and will incorporate our responses into the paper draft to improve its quality and clarity.
>
> > Given this, it seems unclear what are the additionally challenges in this benchmarks compared to for example the AsyncHow dataset mentioned in the paper.
>
> We would like to clarify the key features of Robotouille and how it differs from related benchmarks such as ​​AsyncHow:
> - **Interactive Simulator:** Robotouille has a game engine backend wrapped as an OpenAI Gym environment. This tests the closed-loop planning ability of LLM agents and their ability to recover from failures as it learns about the rules and constraints of the environment. This is different from benchmarks like AsyncHow [1] which do not use an interactive environment.
> - **Language Backend:** Compared to benchmarks like Overcooked-AI [2], Robotouille offers a diverse number of tasks (30) with infinite procedurally generated kitchens, and a language backend that supports LLM agents
> - **Asynchronous Planning:** Compared to benchmarks like VirtualHome [3] and Planbench [4], which have been used to benchmark LLM synchronous planning, Robotouille introduces time delays to benchmark LLM asynchronous planning.
>
> [1] Graph-enhanced Large Language Models in Asynchronous Plan Reasoning. Lin et. al 2024
>
> [2] On the Utility of Learning about Humans for Human-AI Coordination. Carroll et. al 2019
>
> [3] VirtualHome: Simulating Household Activities via Programs. Puig et. al 2018
>
> [4] PlanBench: An Extensible Benchmark for Evaluating Large Language Models on Planning and Reasoning about Change. Valmeekam et. al 2022
>
> > It is unclear from the paper which components of asynchronous planning make it much harder than synchronous planning. A detailed analysis of why “asynchronous” planning is harder is needed to understand in what ways do this benchmark challenges the methods.
> >
> > Which components of asynchronous planning make it much harder than synchronous planning?
>
> We provide a sketch of why asynchronous planning is harder than synchronous planning below by analyzing the complexity of search for both settings.
>
> We can model time-delayed actions as expanding the state space of the MDP with timer variables that track how long such an action has been active for.  Consider an MDP with time-delayed effects, represented as $<\mathcal{S}, \mathcal{A}, \mathcal{T}, \mathcal{R}>$, where:
>
> - State Space $\mathcal{S}$: Each state $s \in \mathcal{S}$ is $s = (\hat{s}_t, H_t)$,
>   - $\hat{s}_t$ represents observable state elements (e.g., objects, predicates).
>   - $H_t$ is a set of timer variables $h \in H_t$, each created by actions with a countdown function $h(x) = d - (x - i)$, where $d$ is a delay constant, and $i$ is the timer’s activation step.
> - Actions $a \in \mathcal{A}$ may introduce new timers
> - Transition Function $T(s, a) = s’ = ( \hat{s}\_{t+1}, H_{t+1})$:
>   - State Update: If any timer expires, $ \hat{s}_{t+1} = \hat{s}_t$  $\cup$ { $\text{predicates}(h) \mid h \in H_t, h(t) = 0$ }
>   - Timer Update: $H\_{t+1} =$ $(H\_t - $ { $h \mid h(t) = 0$ } $)$ $ \cup $ { $ h \mid a \text{ adds delay}$ }
> - Reward Function $R$:
>   - $R(s) = 0$ if $s = g$ (goal), else $R(s) = -1$
>
> The complexity of search for synchronous and asynchronous is:
> 1. Synchronous Case $(d = 0)$: No delays, so the planner operates in $O(|S| + |A|)$.
> 2. Asynchronous Case $( d > 0 )$: Each delay expands the effective state space, yielding $O(|S| \times (d+1)^n + |A|)$ complexity, where $n$ is the number of timers.
>
> Hence the expanded state space requires both a conventional planner or a LLM based planner to reason over a larger range of delayed effects. We are happy to update the paper draft with this simple analysis to show why asynchronous planning is harder.

---

> > ### Author Response · Authors · 2024-11-15
> >
> > > In the failure mode analysis, it is unclear to me what does it mean to attribute failure in terms of uncertainty over the MDP. Additionally, this categorization seems to be irrelevant to the key ingredient of the benchmark: asynchronous planning.
> >
> > We clarify that Robotouille is a benchmark that evaluates LLM decision-making for **both** synchronous and asynchronous settings. Our evaluation of LLM agents in Robotouille surfaced fundamental failures of the agent common to both settings that we capture in the pie chart in Figure 5. The pie chart uses a taxonomy that classifies each LLM agent failures into uncertainty about a component of the MDP: state, action, transition or goals. The failure occurs because the agent is unable to resolve the uncertainty from environment feedback. The piechart was essential in understanding where the bulk of the failures lie for LLM agents to help future research. For instance, the piechart revealed that a frequent failure mode was due to the  'One item at a station' rule which is an uncertainty about transition, i.e. more than one item cannot be placed at a station. This led us to improving the prompt for LLM plan generation to explicitly reason about transitions better, leading to a reduction in failure from 42.9% to 27.8% on a subset of the asynchronous dataset. We believe this pie-chart is a valuable contribution that is lacking in current LLM benchmarks like PlanBench [1], and can help future work in decreasing these failures.
> >
> > [1] PlanBench: An Extensible Benchmark for Evaluating Large Language Models on Planning and Reasoning about Change. Valmeekam et. al 2022
> >
> > > While this paper proposes a new benchmark for certain types of LLM reasoning tasks, it does not include sufficient experimental evaluation and analysis to highlight its main challenges. Specifically, while there are many LLMs as well as LLM-based planning algorithms to tackle planning problems, the paper only experimented with one LLM and two planning algorithms (CoT and ReAct). More baselines are needed to better illustrate the challenges posed by the proposed benchmark.
> >
> > In Table 2 we present results using 4 state-of-the-art LLMs (gpt4-o, gpt-4o-mini, gemini-1.5-flash, claude-3-haiku) on the synchronous and asynchronous datasets. We also further clarify our ReAct baseline to highlight how our prompts make a fair attempt at tackling the synchronous and asynchronous settings.
> >
> > For the synchronous setting, ReAct is few-shot prompted to generate a plan by (1) decomposing a goal into ordered subgoals and (2) tracking the progress of each subgoal sequentially before selecting an action and replanning at the next timestep.
> >
> > For the asynchronous setting, ReAct is few-shot prompted to additionally prioritize asynchronous subgoals when the goal is decomposed into ordered subgoals.
> >
> > We are happy to include two additional baselines: Reflexion [1] and PLaG [2].
> > - Reflexion is a self-refinement on React that makes multiple attempts at the same environment. We expect it to make improvements at the cost of an increased number of actions.
> > - We expect PLaG to perform roughly the same as ReAct since our baseline essentially builds a high-level dependency graph to plan with.
> >
> > [1] Reflexion: Language Agents with Verbal Reinforcement Learning. Shinn et. al 2023
> >
> > [2] Graph-enhanced Large Language Models in Asynchronous Plan Reasoning. Lin et. al 2024
> >
> >
> > > While the paper highlights that the benchmark requires long-horizon planning, the horizon counted by the paper is the number of required “atomic steps”, while the agent is performing high-level actions such as move an object to a place. This size of the graphs (e.g., in Figure 11) seems to be a better measure of the “planning depth”. Given this, it seems unclear what are the additionally challenges in this benchmarks compared to for example the AsyncHow dataset mentioned in the paper.
> >
> > The horizon lengths that we include in Table 3 do indeed refer to the high-level actions that the agent is performing. We refer the reviewer to Appendix A.3 as the prompt includes the actions the agent can take.
> >
> > The dependency graphs like Figure 11 show the task conceptually and do not indicate the resolution at which planning occurs. Each node is a subtask that requires planning to solve that subtask.
> >
> > AsyncHow’s maximum plan horizon length is 9 as we show in Table 1. We also emphasize that AsyncHow’s setting is open-loop since there is no interactive environment for an LLM agent to retry after giving a time estimate. This contrasts to Robotouille where an LLM agent must traverse in an environment and recover from mistakes using observation feedback.

---

> > > ### Author Response · Authors · 2024-11-22
> > >
> > > We have provided initial results below for our new implementation of PLaG (BaG) and Reflexion (with one retry) on Task 1 and 2 of the asynchronous dataset. We will continue these experiments and add them to the paper.
> > >
> > > We initially expected PLaG (BaG) to achieve similar performance to our ReAct baseline since our ReAct baseline constructs an ordered sequence for subgoals to complete while PLaG (BaG) constructs an adjacency list that captures dependencies between subgoals which dictate the order of subgoals. We see this in the results and expect to continue seeing this trend on other tasks.
> > >
> > > We initially expected Reflexion to make improvements on our ReAct baseline since it makes additional attempts at the same environment with reflections from failures in past attempts. While performance strictly increases, we found that these attempts succeeded on the first attempt meaning retrying had no effect. We examined the reasoning in the second attempt and verified that the agent makes use of the reflection which subsequently changes its behavior; however, we note that the agent continuously violates the same environment constraints suggesting that the reflections are not targeting the core agent failure. We expect long context summarization into useful reflections to be difficult because LLMs are ineffective with long context [1]. Increasing the number of attempts could improve performance but this is expensive and does not address the core issue that LLMs are poor at summarizing long context into actionable insights.
> > >
> > > Please let us know if there are any additional questions that we can answer. If you are satisfied with the responses, please consider revising your score to reflect this.
> > >
> > > [1] Lost in the middle: How language models use long contexts. Liu et al. 2023.
> > >
> > > | Task | ReAct | PLaG (BaG) | Reflexion  |
> > > |------|-------|------------|------------|
> > > | 1    | 20.0  | 20.0       | 30.0       |
> > > | 2    | 30.0  | 30.0       | 50.0       |

---

> > > > ### Author Response · Authors · 2024-12-02
> > > >
> > > > We believe our revision and responses have addressed most of your concerns. Summarizing them below
> > > > - We have introduced new baselines, PLaG (BaG) and Reflexion, and shared performance trends on asynchronous tasks in our previous message. PLaG (BaG) matches our ReAct baseline in performance and Reflexion improves performance at the cost of increased token usage through repeated environment runs.
> > > > - We have demonstrated why asynchronous planning is more difficult than synchronous planning through an analysis of the complexity for our MDP and revised the paper to include this analysis. We have additionally augmented our dataset through discussion with reviewer *nvj3* to incorporate stochasticity as strictly harder versions of the synchronous and asynchronous settings to further challenge methods.
> > > > - We have demonstrated that Robotouille is significantly different from AsyncHow in its use of an interactive environment. Additionally, AsyncHow does not tackle stochasticity.
> > > >
> > > > As the author-reviewer discussion period ends soon, we would be happy to address any further questions during this time and hope you consider increasing the score.

---

> > > > > ### Author Response · Authors · 2024-12-04
> > > > >
> > > > > We present the final baseline results for all 10 tasks in the asynchronous dataset, evaluated over 10 seeds per task. We make the following observations:
> > > > >
> > > > > - Compared to the ReAct baseline, PLaG (BaG) achieves a total performance improvement of 2%, while Reflexion achieves a 6% increase
> > > > > - Using an explicit dependency graph during prompting provides slight performance boosts as demonstrated by PLaG (BaG)
> > > > > - Despite reflecting on past experience and retrying a task, Reflexion only achieves 17% on the asynchronous dataset demonstrating the challenge of feedback incorporation in our benchmark
> > > > > - All methods fail on the last 4 tasks due to their difficulty
> > > > >
> > > > > We will include these new results in our paper to illustrate the challenges our benchmark presents and emphasize that effective feedback incorporation is a critical issue for improving LLMs' performance in both synchronous and asynchronous planning.
> > > > > | Task | ReAct | PLaG (BaG) | Reflexion  |
> > > > > |------|-------|------------|------------|
> > > > > | 1    | 20.0  | 20.0       | 30.0       |
> > > > > | 2    | 30.0  | 30.0       | 50.0       |
> > > > > | 3    | 40.0  | 20.0       | 40.0       |
> > > > > | 4    | 10.0  | 30.0       | 10.0       |
> > > > > | 5    | 0.00  | 10.0       | 10.0       |
> > > > > | 6    | 10.0  | 20.0       | 30.0       |
> > > > > | 7    | 0.00  | 0.00       | 0.00       |
> > > > > | 8    | 0.00  | 0.00       | 0.00       |
> > > > > | 9    | 0.00  | 0.00       | 0.00       |
> > > > > | 10   | 0.00  | 0.00       | 0.00       |

---

### Meta-Review · Area_Chair_iazf · 2024-12-29

**Metareview:**

The paper introduces Robotouille, an asynchronous planning benchmark, designed to evaluate how well agents can handle long-horizon tasks, time delays, interruptions, and collaboration. The benchmark found that current LLM agents struggle with asynchronous tasks, with ReAct (gpt-4o) achieving 47% on synchronous tasks but only 11% on asynchronous tasks. The authors also provide a failure mode analysis, showing that LLM agents need to improve in incorporating long-horizon feedback and self-auditing their reasoning. The authors demonstrate that the complexity of asynchronous planning increases due to time delays expanding the state space.

An an asynchronous planning benchmark would significantly propel the current state of the art in the AI Agents, and the benefit to the field outweighs the paper's current weaknesses.

**Additional Comments On Reviewer Discussion:**

The reviews of the paper were mixed, with reviewers raising concerns about the lack of sufficient experimental evaluation and analysis, prompt design, and comparison between synchronous and asynchronous tasks. The authors made a strong rebuttal and significantly improved the paper, addressing these concerns by adding new baselines, and sharing performance trends on asynchronous tasks. They also clarified their prompt design and removed the comparison between the synchronous and asynchronous datasets, focusing on the claim that asynchronous settings are strictly harder. Additionally, they augmented the dataset with stochasticity and explained why Robotouille differs from the AsyncHow dataset.

---

### Decision · Program_Chairs · 2025-01-22

Accept (Poster)